# HiPO: Self-Hint Policy Optimization for RLVR

**Qiyuan Deng**[1], **Kehai Chen**[1]*, **Min Zhang**[1], **Zhongwen Xu**

[1]Harbin Institute of Technology, Shenzhen, China
{chenkehai, zhangmin2021}@hit.edu.cn
23S151120@stu.hit.edu.cn

## Abstract

Reinforcement Learning from Verifiable Rewards (RLVR) is a promising method for enhancing the complex problem-solving abilities of large language models (LLMs). This is particularly evident in domains requiring long-horizon reasoning and precise execution, such as solving complex mathematical problems where solutions hinge on a fragile sequence of tool-based actions. However, current approaches are often crippled by two interconnected issues: the near-miss problem, where sparse rewards nullify the learning signal for almost-correct attempts, and the resulting exploration stagnation, which prevents the model from discovering better solutions. To address these challenges, we introduce HiPO (Hint-guided Policy Optimization), a novel RLVR framework that enables the agent to learn from its own rare successes. Our core insight is to capture an occasional successful trajectory within a training batch and repurpose its initial correct steps as an on-policy "hint". This process transforms a single, stochastically-found success into a dense contrastive learning signal, effectively allowing the model to teach itself how to overcome the near-miss problem and break exploration stagnation. On a challenging suite of five mathematical reasoning benchmarks, HiPO improves the average avg@32 by +5.0 percentage points (pp) over the strong GRPO baseline. This improvement is driven by substantial absolute point gains on challenging datasets, including +10.3 pp on CMIMC 2025, +4.9 pp on BRUMO 2025, +4.6 pp on AIME 2024, and +3.1 pp on AIME 2025. Furthermore, HiPO demonstrates a new exploration paradigm, repurposing rare successes into reusable guidance to significantly accelerate skill acquisition for complex tasks, establishing a more efficient and scalable path for models to autonomously master intricate reasoning.

## 1 Introduction

A central ambition in artificial intelligence is to create agents capable of acquiring complex reasoning skills autonomously, moving beyond the reliance on curated expert data (DeepSeek, 2025; Lambert et al., 2024). Reinforcement Learning from Verifiable Rewards (RLVR) marks a promising step towards this vision, enabling Large Language Models (LLMs) to learn from outcome-based feedback in domains like advanced mathematics (Shao et al., 2024; DeepSeek, 2025) . However, this pursuit confronts a fundamental paradox rooted in the very nature of complex problem-solving. Success often depends on a fragile sequence of self-reflection (Renze & Guven, 2024) and tool-integrated reasoning (Feng et al., 2025; Li et al., 2025b), making a correct solution an exceptionally rare event. This fragility exposes a critical flaw in current RLVR frameworks: a *brittleness of the learning signal*.

This signal brittleness manifests in two interconnected challenges. The first is the *near-miss problem*: a nearly-perfect trajectory receives the same sparse, negative reward as a complete failure, thereby penalizing correct intermediate reasoning steps through flawed credit assignment. Consequently, this leads to the second, more debilitating challenge: *exploration stagnation*. The consistent punishment for near-misses disincentivizes any deviation from simple, suboptimal strategies, preventing the very breakthroughs in capability the learning process is meant to foster (Cui et al., 2025; An et al., 2025; Cheng et al., 2025). This is further exacerbated by *signal collapse*, a scenario where

---

*Corresponding author.

uniform rewards within a training batch cause the policy gradient to vanish entirely (Yu et al., 2025; Xu & Ding, 2025). In essence, the agent cannot learn from its successes if those successes are statistically impossible to discover.

To make this exploration challenge concrete, consider a motivating example. As shown in Figure 1, a baseline model without guidance struggles to generate correct trajectories. However, when provided with a partial correct solution as a "hint", its performance improves dramatically, transforming the distribution from near-certain failure to high-probability success. This stark contrast suggests the primary bottleneck is not the model's intrinsic reasoning ability, but the *discovery* of a valid reasoning path. This begs the critical question: can a model learn to provide these crucial hints for itself, *bootstrapping* its own learning from its own rare successes?

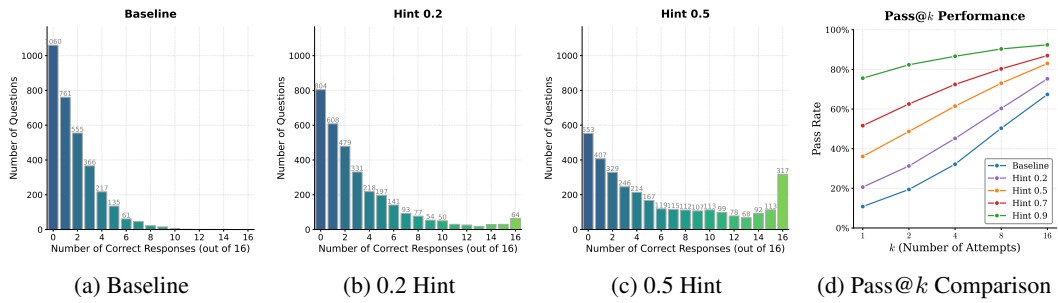

| (a) Baseline | (b) 0.2 Hint | (c) 0.5 Hint | (d) Pass@$k$ Comparison |

Figure 1: (a-c) show as the hint ratio (the proportion of the trajectory as a prefix) increases, the correct solutions improves. (d) compares the pass@k performance for different hint ratios.

To answer this question, we introduce Hint-guided Policy Optimization (HiPO), a framework that operationalizes a new paradigm: **Endogenous Self-Hint**. The core insight is that a single, stochastically-found success is not merely an endpoint to be rewarded, but a rich source of curriculum to be exploited. HiPO captures these rare successful trajectories and repurposes their initial correct steps as on-policy "hints". This process transforms the intractable task of *end-to-end discovery* into a more manageable one of *guided completion*. By juxtaposing the policy's unaided exploration with its hint-guided exploration, HiPO forges a dense and powerful contrastive learning signal from sparse rewards. It directly resolves the near-miss problem by rewarding valid reasoning prefixes, and by providing a trustworthy path forward, it shatters exploration stagnation, guiding the policy toward mastering more complex reasoning patterns.

Our main contributions are:

- We propose HiPO, a new framework that materializes the concept of endogenous self-hint. It leverages policy's own rare successes as on-policy hints to create a robust, contrastive learning signal, converting sparse rewards into a dense, self-generated curriculum.

- HiPO significantly outperforms a strong GRPO baseline on five challenging mathematical reasoning benchmarks, achieving a +5.0 pp average improvement in avg@32. The gains are particularly substantial on difficult datasets, including CMIMC 2025 (+10.3 pp) (Balunović et al., 2025), BRUMO 2025 (+4.9 pp) (Balunović et al., 2025), and AIME 2024 (+4.6 pp).

- We provide empirical evidence that HiPO successfully counteracts exploration stagnation. Through analysis of training dynamics, we show that it maintains higher policy entropy and fosters longer, more complex tool-use sequences, confirming that our self-hint mechanism enables more sophisticated reasoning.

These results highlight HiPO as an efficient and scalable paradigm for autonomous learning. By enabling models to repurpose their own successes into reusable guidance, our work provides a bootstrapped pathway to mastering complex tasks, accelerating progress towards more capable and independent AI systems.

## 2 RELATED WORK

A prominent baseline within RLVR (Lambert et al., 2024; DeepSeek, 2025) is Group Relative Policy Optimization (GRPO) (Shao et al., 2024), which simplifies policy optimization by forgoing a critic network and instead normalizing rewards across a group of concurrently generated responses.

However, this group-based approach is crippled when rewards are sparse; if all responses in a training batch fail, the resulting advantage is zero, nullifying the learning signal. To mitigate this, methods like DAPO (Yu et al., 2025) enhance GRPO with engineering treatments such as dynamic sampling, which forces the generation process to continue until a non-zero advantage is found. Despite these improvements, such approaches are fundamentally reactive; they can only refine the learning signal if at least one successful trajectory is stochastically discovered within a group, failing to address the core exploration problem when success is exceptionally rare.

Moving beyond purely algorithmic modifications, another line of work injects external guidance to scaffold learning. QuestA (Li et al., 2025a) and StepHint (Zhang et al., 2025), for instance, augment prompts with partial solutions or stepwise hints derived from *stronger teacher models* or *existing datasets* of OpenR1-Math-220K (OpenR1, 2025). While effective at creating a denser reward signal, this reliance on external, *off-policy* guidance creates a dependency on pre-existing knowledge and does not represent autonomous learning from the agent's own experience. Our work, HiPO, provides a novel synthesis to overcome these limitations. It addresses the sparse reward problem like hint-based methods, but does so by creating hints that are both *endogenous* and *on-policy*. By capturing a rare success within a training batch and repurposing its initial correct steps as a hint for the entire group, HiPO enables the model to teach itself, transforming a single stochastic success into a dense, reusable learning signal without depending on external teacher models.

## 3 BACKGROUND

### 3.1 POLICY GRADIENT IN RLVR

The optimization of RLVR is fundamentally grounded in the policy gradient theorem from reinforcement learning (Sutton & Barto, 2018). We can formalize the sequential problem-solving process as a finite-horizon Markov Decision Process (MDP) (Puterman, 1990). For a given prompt $P$, serves as the initial state $s_0$, the model autoregressively generates a trajectory $\tau = (o_0, o_1, \ldots, o_{T-1})$, a sequence of tokens from a vocabulary $\mathcal{V}$. At each step $t$, the model's policy $\pi_\theta$ defines a probability distribution over the next token $o_t$, conditioned on the current state $s_t$.

The policy $\pi_\theta$, parameterized by the model's weights $\theta$, is the language model itself. In RLVR, a sparse reward $R(\tau)$ is typically assigned only at the end of a generated trajectory $\tau$. The objective is to adjust the parameters $\theta$ to maximize the expected reward over all possible trajectories:

$$J(\theta) = \mathbb{E}_{\tau \sim \pi_\theta}[R(\tau)]. \tag{1}$$

The policy gradient is optimized by computing its gradient with respect to $\theta$, which can be estimated by sampling:

$$\nabla_\theta J(\theta) = \mathbb{E}_{\tau \sim \pi_\theta}[R(\tau)\nabla_\theta \log \pi_\theta(\tau)]. \tag{2}$$

To reduce the high variance of this estimator, a state-dependent baseline $b(P)$ is subtracted from the reward $R(\tau)$ to yield a lower-variance advantage estimate, $A(\tau) = R(\tau) - b(P)$. This results in the final policy gradient update rule:

$$\nabla_\theta J(\theta) = \mathbb{E}_{\tau \sim \pi_\theta}[A(\tau)\nabla_\theta \log \pi_\theta(\tau)]. \tag{3}$$

This rule guides the model to favor trajectories with positive advantages (better than average) and avoid those with negative ones, making a meaningful advantage signal crucial for effective learning.

### 3.2 GROUP RELATIVE POLICY OPTIMIZATION

Group Relative Policy Optimization (GRPO) is a prominent RLVR algorithm that forgoes a learned critic network, which is a key component of the PPO algorithm (Schulman et al., 2017). Instead, it constructs an empirical, on-the-fly advantage signal by generating a group of $n$ trajectories $\mathcal{T}_j = \{\tau_{j,1}, \ldots, \tau_{j,n}\}$ for each prompt $P_j$.

To optimize, GRPO maximizes an objective function based on an importance sampling ratio. For a given trajectory $\tau_{j,i} \in \mathcal{T}_j$, we define the probability ratio for each token $o_{j,i,t}$ at timestep $t$ as:

$$r_t^{(j,i)}(\theta) = \frac{\pi_\theta(o_{j,i,t}|P_j, o_{j,i,<t})}{\pi_{\theta_{\text{old}}}(o_{j,i,t}|P_j, o_{j,i,<t})}, \tag{4}$$

where $o_{j,i,<t}$ represents the sequence of tokens. The objective function is defined as:

$$\mathcal{J}_{\text{GRPO}}(\theta) = \mathbb{E}_{P_j, \mathcal{T}_j} \left[ \frac{1}{n} \sum_{i=1}^{n} \frac{1}{|\tau_{j,i}|} \sum_{t=1}^{|\tau_{j,i}|} \min \left( r_t^{(j,i)}(\theta) \hat{A}_{j,i}, \text{clip} \left( r_t^{(j,i)}(\theta), 1-\epsilon, 1+\epsilon \right) \hat{A}_{j,i} \right) \right]. \tag{5}$$

GRPO's core lies in its advantage normalization, where a single advantage $\hat{A}_{j,i}$ is computed for each trajectory $\tau_{j,i}$ and applied uniformly across all its tokens:

$$\hat{A}_{j,i} = \frac{R(\tau_{j,i}) - \mu_{\mathcal{T}_j}}{\sigma_{\mathcal{T}_j} + \epsilon_{\text{stable}}}. \tag{6}$$

where $\mu_{\mathcal{T}_j} = \frac{1}{n} \sum_{k=1}^{n} R(\tau_{j,k})$ is the mean reward of the group, and $\sigma_{\mathcal{T}_j}$ is its standard deviation. The policy is then updated using this uniform advantage signal across all time steps of the trajectory.

However, while this design is computationally efficient, its reliance on a single outcome reward introduces critical flaws. The first, and for our purposes the most detrimental, is the problem of credit misassignment for fragile trajectories. In the context of complex mathematical reasoning, a trajectory is often "fragile"; a single error can invalidate a long sequence of correct logical steps, thereby resulting in a minimal or zero final reward. Under this reward structure, the negative feedback derived from this outcome is uniformly distributed across all tokens in the sequence. Consequently, the vast majority of correct and valuable reasoning steps are unduly penalized, which actively impedes the model's acquisition of the long-form logic required for challenging problems.

Furthermore, the reliance on the formulation in Equation 6 gives rise to another critical flaw: the signal collapse problem (Yu et al., 2025; Xu & Ding, 2025). This issue materializes in what we term a "null-signal group", a scenario wherein all trajectories happen to share the same reward. In such cases, both the numerator ($R(\tau_{j,i}) - \mu_{\mathcal{T}_j}$) and the denominator ($\sigma_{\mathcal{T}_j}$) of the advantage calculation nullify. This inevitably results in $\hat{A}_{j,i} = 0$ for every sample, which constitutes a catastrophic loss of the learning signal that leads to the stagnation of exploration.

## 4 THE HiPO FRAMEWORK

### 4.1 CORE MECHANISM: ON-POLICY HINT

The core mechanism of HiPO constructs a contrastive learning signal by juxtaposing the policy's unaided exploration with its hint-guided exploration. HiPO enriches low-signal batches by using rare successes to generate high-signal replacements for completely unlearnable groups. This process unfolds in two phases for each prompt $P_j$ within a given mini-batch.

First, the policy attempts to solve the prompt without assistance, generating a standard Group of $n$ trajectories, $\mathcal{T}_{\text{orig},j}$. This group faithfully reflects the model's current capabilities.

$$\mathcal{T}_{\text{orig},j} = \{\tau_1, \dots, \tau_n\} \quad \text{where} \quad \tau_i \sim \pi_\theta(\cdot|P_j). \tag{7}$$

From this initial generation, we identify specific low-performing groups as "Near-miss Groups" $\mathcal{T}_{\text{near-miss},j}$, which are defined as those where the success rate of rollouts falls below half, and "Unlearnable Groups" $\mathcal{T}_{\text{null-signal},j}$, where the reward variance is zero. The upper portion of Figure 2 visually represents a $\mathcal{T}_{\text{near-miss},j}$, characterized by a high proportion of failed rollouts (red) and a scarcity of successful ones (green), which typifies the challenge in complex reasoning tasks.

To counteract this, we employ a guided exploration strategy. First, an on-policy hint pool is constructed from the set of all successful trajectories within the near-miss group: $\mathcal{H}_{\text{pool},j} = \{\tau \in \mathcal{T}_{\text{near-miss},j} \mid R(\tau) = 1\}$. The core of this strategy is to capitalize on these rare successes to generate

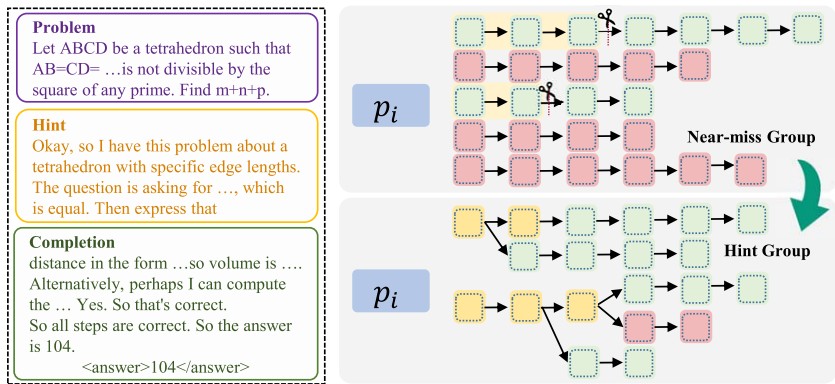

Figure 2: An illustration of the HiPO core mechanism.

a new, high-signal Hinted Group, $\mathcal{T}_{\text{hint},j}$. To ensure diversity within this new group, each trajectory is generated using a unique hint. This hint is created via a two-stage sampling process: first, a source trajectory $\tau_{\text{source}}$ is uniformly sampled from $\mathcal{H}_{\text{pool},j}$. Second, a prefix ratio $p$ is sampled from discrete values in the range $[0.05, 0.45]$ with a step of $0.05$ to determine a hint length $k = \lfloor p \cdot |\tau_{\text{source}}| \rfloor$. This procedure yields a unique hint, $H_{j,i} = \text{Prefix}(\tau_{\text{source}}, k)$, which represents the initial steps of a successful attempt. As illustrated in Figure 2, this process corresponds to randomly truncating a successful trajectory. The final trajectory is then generated conditioned on the hint being appended to the original prompt: $\tau'_i \sim \pi_\theta(\cdot | P_j \oplus H_{j,i})$. A detailed discussion of the hint generation strategy is provided in Appendix D

The theoretical underpinning of this method is its principled resolution of sample inefficiency. A naive alternative for improving sample efficiency might involve augmenting the training data with off-policy expert trajectories drawn from a distribution $\pi_E$. This approach, however, suffers from two fundamental drawbacks. The most critical is a practical one: the requisite high-quality expert data is often prohibitively expensive or simply unavailable, as its acquisition demands either extensive manual annotation or a pre-existing, superior "teacher" model. This reliance on external supervision contrasts sharply with self-sufficient, bootstrapping methods like our own. Secondly, even if such data were accessible, its introduction engenders a significant distributional mismatch between the agent's policy $\pi_\theta$ and the expert policy $\pi_E$. This discrepancy is a well-documented cause of training instability, hindering reliable convergence by destabilizing the gradient estimator.

---

**Algorithm 1** The Hint Mechanism of HiPO

**Input:** A prompt $P$, current policy $\pi_\theta$, group size $n$.
**Output:** An augmented group of trajectories.
1: **procedure** GENERATEAUGMENTEDGROUP($P, \pi_\theta, n$)
2:     $\mathcal{T}_{\text{orig}} \leftarrow$ Generate $n$ trajectories for $P$ using policy $\pi_\theta$.
3:     $\mathcal{H}_{\text{pool}} \leftarrow \{\tau \in \mathcal{T}_{\text{orig}} \mid R(\tau) = 1\}$.
4:     **IF** $0 < |\mathcal{H}_{\text{pool}}| < n/2$ **then**
5:         $\mathcal{T}_{\text{hint}} \leftarrow \emptyset$.
6:         **FOR** $k = 1, \ldots, n$ **do**
7:             $H \leftarrow \text{ExtractRandomHint}(\mathcal{H}_{\text{pool}})$.
8:             Generate $\tau'_{\text{new}}$ from $P \oplus H$ and add to $\mathcal{T}_{\text{hint}}$.
9:         **end FOR**
10:        **RETURN** $\mathcal{T}_{\text{hint}}$
11:     **end IF**
12: **end procedure**

---

Ultimately, the policy update leverages data from both the Original and Hinted groups. This transforms a potentially degenerate learning signal into a well-posed, contrastive optimization problem. This self-teaching process, formalized in Algorithm 1, is strategically activated to create this contrast

precisely when it is most needed. The hint-generation phase is triggered for what we term "Near-miss Groups": those where the success rate is greater than zero but falls below half. This targets the critical scenarios where initial successes are present but fragile, representing moments where the model is on the cusp of a breakthrough but still requires guidance. The resulting juxtaposition of the low-success-rate Original Group and the high-success-rate Hinted Group creates a highly informative advantage signal: a positive advantage reinforces the completion of nascent but promising reasoning paths, as demonstrated in the Hinted Group, while a negative advantage penalizes identifiable failure modes that persist even with guidance. HiPO thereby offers a more principled and efficient path toward mastering complex reasoning.

## 4.2 How HiPO Creates Dense Learning Signals from Sparse Rewards

We frame complex reasoning under sparse rewards as a problem of leveraging the structure of successful trajectories. However, frequent failures often lead to *signal collapse*, which is a scenario where zero reward variance within a group nullifies the advantage estimate and halts learning. To overcome this, HiPO extracts intermediate states from rare successes to serve as "hints" for initiating new, guided trajectories. This approach transforms the difficult task of end-to-end discovery into the more manageable one of completing a partially successful reasoning path, generating a dense learning signal from both unaided and hint-guided rollouts.

HiPO's refines the exploration strategy by modifying the initial state distribution for a subset of rollouts. We define the set of successful trajectories from the unaided exploration phase as $\mathcal{T}_{\text{near-miss}}^+ \triangleq \{\tau \in \mathcal{T}_{\text{near-miss}} \mid R(\tau) = 1\}$. We define a hint $H$ as an intermediate state $s_k$ within a successful trajectory, where $s_k \in \tau$ for some $\tau \in \mathcal{T}_{\text{near-miss}}^+$. Trajectories in the Hinted Group, $\mathcal{T}_{\text{hint}}$, are then generated by sampling from the model $\pi_\theta$ conditioned on the concatenation of an original prompt $P_j$ and a hint $H$. This process is formally expressed as $\tau' \sim \pi_\theta(\cdot|P_j \oplus H)$, where $H$ is a state drawn from a successful trajectory. This acts as a form of value-guided exploration. While the optimal value function $V^*$ is unknown, states from empirically successful trajectories serve as effective proxies for high-value states. This method also prevents signal collapse by diversifying rewards within a batch, which averts the vanishing advantage estimate $\hat{A}_\tau$ from identically-rewarded trajectories. By ensuring signal diversity, HiPO avoids the catastrophic loss of learning signal and the subsequent stagnation of exploration.

To counteract this, HiPO implements a strategic batch replacement. Let $\mathcal{B}_{\text{orig}}$ denote original groups generated in the unaided exploration. The method sources successful trajectories from $\mathcal{T}_{\text{near-miss}}$ within this batch to generate $\mathcal{T}_{\text{hint}}$. These new groups then strategically replace the $\mathcal{T}_{\text{null-signal}}$ that offer no learning gradient. This process is formalized as:

$$\mathcal{B}_{\text{HiPO}} \triangleq (\mathcal{B}_{\text{orig}} \setminus \mathcal{T}_{\text{null-signal}}) \cup \mathcal{T}_{\text{hint}}. \tag{8}$$

The gradient estimator over this optimized batch is:

$$\hat{g}_{\text{HiPO}} = \mathbb{E}_{\tau \sim \mathcal{B}_{\text{HiPO}}} \left[ \sum_{t=1}^{|\tau|} \nabla_\theta \min \left( r_t^{(\tau)}(\theta) \hat{A}_\tau, \text{clip}\left( r_t^{(\tau)}(\theta), 1 - \epsilon, 1 + \epsilon \right) \hat{A}_\tau \right) \right], \tag{9}$$

where $\hat{A}_\tau$ is the advantage estimate and $r_t^{(\tau)}$ is the importance sampling ratio. For any trajectory $\tau$ in the augmented batch $B_{\text{HiPO}}$, its advantage is in its respective group. By replacing, this formulation decomposes the learning objective into a structured set of signals. The gradient is shaped by four distinct, high-value scenarios. Rare, successful trajectories remaining in $\mathcal{B}_{\text{orig}}$ receive a strong positive signal, anchoring the policy. In contrast, failed trajectories remaining in $\mathcal{B}_{\text{orig}}$ are penalized, though their signal may contain "near-miss" paths. Successful trajectories in $\mathcal{T}_{\text{hint}}$ form the core learning signal by salvaging near-misses from high-value states. Most critically, failed trajectories in $\mathcal{T}_{\text{hint}}$ provide a clear, high-quality negative signal, precisely penalizing deviations from a known-good path. By disentangling the learning signal, HiPO effectively transforms the learning problem from one of sparse rewards and catastrophic signal loss to a structured task of guided completion.

## 5 EXPERIMENTS

### 5.1 EXPERIMENTAL SETTING

We train our model on the DAPO dataset (Yu et al., 2025), a challenging collection of 17K mathematical problems with integer answers sourced from the AoPS community. For evaluation, we test our model across a comprehensive suite of recent mathematics competitions, including the AIME 2024, AIME 2025, BRUMO 2025 (Balunović et al., 2025), HMMT Feb 2025 (Balunović et al., 2025), CMIMC 2025 (Balunović et al., 2025), and Apex 2025 (Balunović et al., 2025). Performance is measured by the average accuracy across 32 generated samples per problem (avg@32), along with the majority vote accuracy (maj@32) and pass rate (pass@32).

Our approach is based on the Qwen3-8B (Qwen, 2025) model, which we trained our model using the VeRL (Sheng et al., 2024) and using the ReTool framework (Feng et al., 2025; Lin & Xu, 2025). ReTool is a reinforcement learning paradigm that teaches the LLM to utilize a Python code interpreter. It learns from outcome-based feedback over multi-turn interactions, with a maximum of eight turns per math prompt.

Key training hyperparameters include a learning rate of 1e-6, a batch size of 96, a mini-batch size of 12, and a maximum response length of 16K tokens. We adopt the Clip-Higher strategy (Yu et al., 2025), setting $\varepsilon_{low}$ to 0.2 and $\varepsilon_{high}$ to 0.28. At each training step, the policy is updated using rewards calculated from 16 sampled responses per prompt. For evaluation, a maximum response length of 32K tokens is used.

### 5.2 EMPIRICAL COMPARISON WITH BASELINE

Table 1: Comparison of HiPO, GRPO and DAPO on five benchmarks using avg@32, pass@32, and maj@32. Averages are shown in the last column. Bold indicates the better-performing method for each metric. For the DAPO, the generation batch size is set to 192, and the maximum number of generation batches is 2.

| Model | AIME 2024 | | | AIME 2025 | | | BRUMO 2025 | | |
| | avg@32 | pass@32 | maj@32 | avg@32 | pass@32 | maj@32 | avg@32 | pass@32 | maj@32 |
|---|---|---|---|---|---|---|---|---|---|
| Qwen3-8B | 54.7 | 85.5 | 60.0 | 47.6 | 84.9 | 63.3 | 30.3 | 55.5 | 56.7 |
| GRPO | 72.1 | **91.7** | 60.0 | 63.0 | 87.1 | 70.0 | 41.7 | 52.1 | 53.3 |
| DAPO | 76.0 | 89.5 | 60 | 63.7 | 87.9 | 70 | **47.8** | 56.6 | 63.3 |
| HiPO | **76.7** | 89.8 | **63.3** | **66.1** | **88.3** | **76.7** | 46.6 | **56.6** | **63.3** |

| Model | HMMT 2025 | | | CMIMC 2025 | | | Average | | |
| | avg@32 | pass@32 | maj@32 | avg@32 | pass@32 | maj@32 | avg@32 | pass@32 | maj@32 |
|---|---|---|---|---|---|---|---|---|---|
| Qwen3-8B | 14.0 | 38.7 | 40.0 | 37.0 | 75.0 | 60.0 | 36.7 | 67.9 | 56.0 |
| GRPO | 28.6 | 48.2 | **46.7** | 43.5 | 75.0 | 55.0 | 49.8 | 70.8 | 57.0 |
| DAPO | **31.4** | 48.4 | 43.3 | 49.9 | 80.0 | 52.5 | 53.7 | 72.4 | 57.8 |
| HiPO | 30.8 | **49.2** | 40.0 | **53.8** | **80.0** | **65.0** | **54.8** | **72.8** | **61.7** |

Our experiments demonstrate that HiPO consistently outperforms the GRPO and DAPO (Yu et al., 2025) baseline on challenging mathematical reasoning benchmarks. As shown in Table 1, HiPO achieves superior aggregate scores on all primary metrics. Specifically, it obtains an avg@32 of 54.8 compared to GRPO's 49.8, representing a significant improvement of +5.0 pp. This aggregate strength is underscored by a notable consistency, as HiPO outperforms GRPO on the avg@32 metric across all five benchmarks. Notably, DAPO's dynamic sampling incurs significant computational overhead, as each training step requires processing a candidate batch of prompts that is a multiple of the size of the batch ultimately used for the gradient update. As shown in Appendix B, DAPO's dynamic sampling incurs significant computational overhead, consuming approximately 4× the prompt volume of HiPO to achieve comparable performance.

The performance disparity is most pronounced on the CMIMC 2025 dataset, where HiPO achieves a substantial +10.3 pp gain. Further significant improvements are observed on BRUMO 2025 (+4.9 pp) and AIME 2024 (+4.6 pp), underscoring the robustness of HiPO's advantages. While

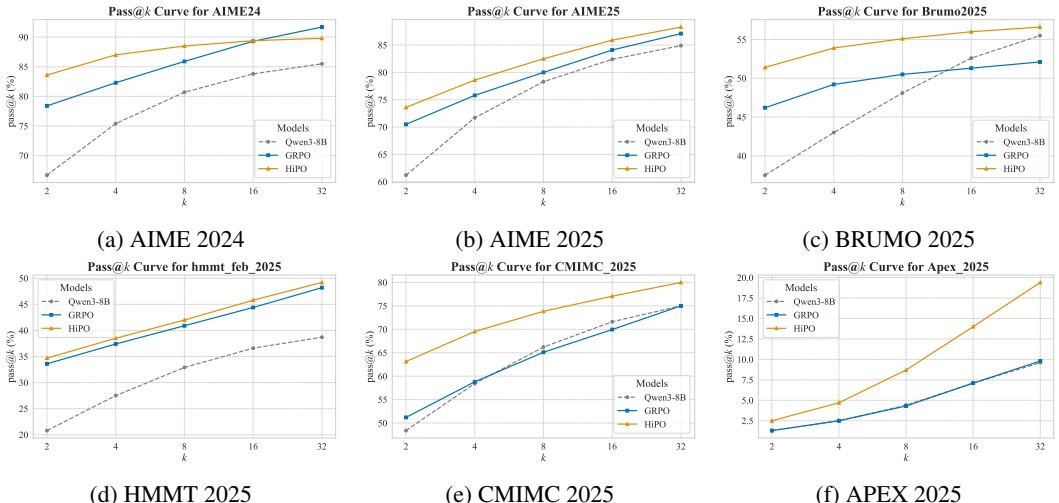

Figure 3: Pass@k curves for different benchmarks.

GRPO is competitive on certain metrics, HiPO's consistent and significant lead, particularly on the most difficult datasets, suggests it learns more robust and generalizable reasoning pathways.

These findings are further supported by the pass@k performance (Chen et al., 2021) curves in Figure 3. The plots generally indicate an advantage for HiPO in sample efficiency, as it often achieves a higher pass rate at lower values of $k$. This trend is most pronounced on the exceptionally challenging Apex 2025 dataset, where the performance gap between HiPO and GRPO widens dramatically as $k$ increases. On this benchmark, HiPO's pass@32 score is nearly double that of the baseline, which suggests that the benefits of its signal reshaping mechanism are particularly salient on problems where successful trajectories are rare. While the performance of both methods converges at higher values of $k$ on some benchmarks, such as AIME 2024, HiPO maintains a clear and consistent performance lead across most other datasets, including Brumo 2025 and AIME 2025, highlighting the potential robustness of our approach. Furthermore, we empirically demonstrate HiPO's robustness in ultra-sparse reward regimes by addressing the cold start problem on the hardest subset of the Omni-Math dataset (Gao et al., 2024) in Appendix C.

## 5.3 TRAINING DYNAMICS

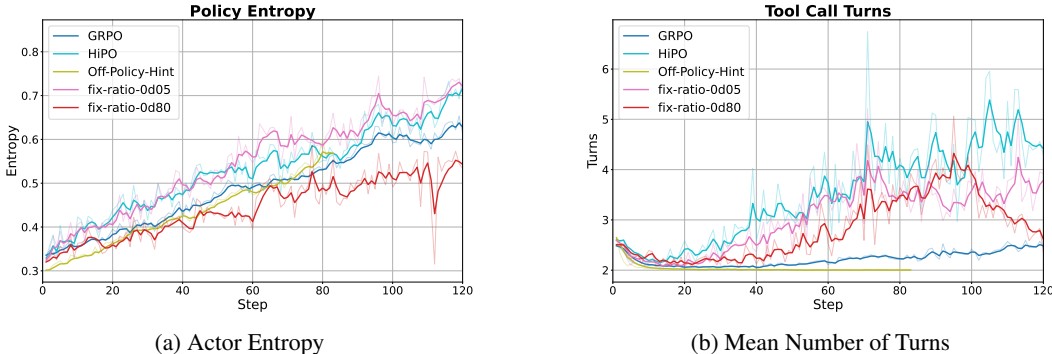

Figure 4: Training dynamics of actor entropy and mean number of turns.

To understand the mechanisms driving HiPO's superior performance, we analyze two key metrics throughout the training process: policy entropy and the average tool-use turns. As plotted in Figure 4. Figure 4a tracks the policy entropy, a measure of policy stochasticity. The plot shows that HiPO (cyan) consistently maintains a higher level of entropy than GRPO (blue). This provides empirical evidence that HiPO successfully mitigates the problem of exploration stagnation (Cui

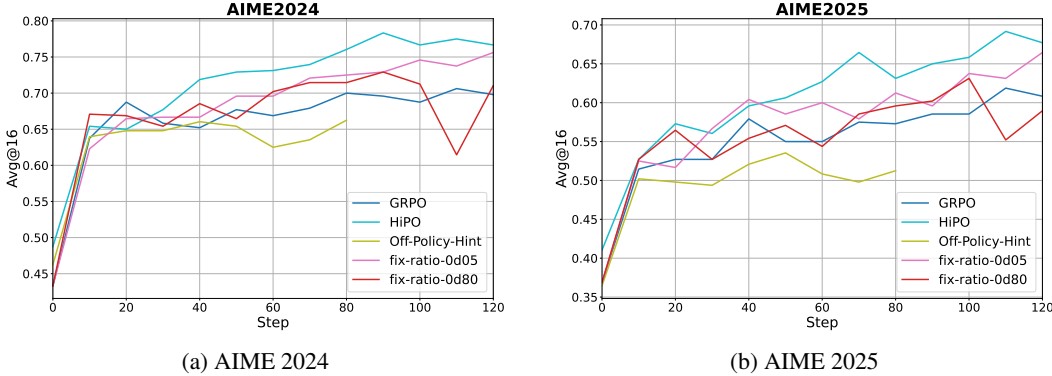

Figure 5: **Ablation on hint ratios.** (a-b) Training curves on AIME 2024 and AIME 2025 benchmarks compared across different hint strategies. (c) The evolution of policy entropy during training. (d) The average number of tool-use turns throughout the training process.

et al., 2025; Wang et al., 2025). GRPO's penalization of near-miss trajectories suppresses exploration, causing the policy to converge on suboptimal, low-diversity strategies. In contrast, HiPO's on-policy hinting mechanism stabilizes the learning signal, promoting the exploration of diverse reasoning pathways and preventing premature policy collapse. This sustained diversity is crucial for discovering more effective problem-solving strategies. The case study in Appendix F provides a concrete example of this behavioral difference. It shows that the HiPO agent identifies an alternative algebraic structure to simplify the problem, whereas the GRPO agent follows a more direct but computationally complex path that leads to an incorrect result.

Complementing the entropy analysis, Figure 4b illustrates the model's capacity for complex, multi-step reasoning by plotting the tool-use turns. A stark divergence emerges: the average number of turns for HiPO trends significantly upwards, indicating that the model learns to engage in longer interactions with the code interpreter. Conversely, the GRPO baseline struggles to increase, showing minimal growth. This suggests that GRPO's flawed credit assignment makes longer reasoning chains brittle and risky, incentivizing the model to adopt simplistic strategies. HiPO, by providing a scaffold for exploration, enables the model to successfully learn and execute the longer, more complex reasoning chains necessary to solve challenging mathematical problems. Taken together, these dynamics provide compelling evidence that HiPO directly counteracts exploration stagnation. By preserving policy diversity, it enables the model to discover and master the longer, more complex reasoning chains necessary for advanced problem-solving.

### 5.4 ABLATION STUDY: ANALYSIS OF HINT RATIO

To validate the necessity of our dynamic hint strategy, we evaluate a low-ratio variant with $p = 0.05$ to represent minimal guidance, and a high-ratio variant with $p = 0.80$ to simulate excessive guidance. Figure 5 confirms that HiPO consistently outperforms static hint strategies. The mechanisms driving these results are revealed in the training dynamics. The high-ratio variant ($p = 0.8$) exhibits low entropy and minimal tool-use turns. This indicates that excessive guidance restricts exploration, trapping the model in local sub-optima where it relies on simple completion rather than learning robust reasoning. Conversely, the low-ratio variant ($p = 0.05$) maintains high entropy but fails to increase tool usage. This suggests that while the model is actively exploring, the lack of sufficient scaffolding prevents it from effectively discovering complex, superior trajectories. HiPO achieves the highest tool-use frequency while maintaining healthy entropy, demonstrating that our dynamic strategy successfully strikes a critical balance between exploration and exploitation, guiding the model toward sophisticated solutions without sacrificing diversity.

### 5.5 ABLATION STUDY: ON-POLICY VS. OFF-POLICY HINTS

To determine whether the efficacy of HiPO stems from the content of the hints or the on-policy nature of their generation, we compare HiPO against an "Off-Policy Hint" baseline. We simulates

a standard teacher-student distillation by collecting successful trajectories from the base model, extract the first 20% of tokens as static hints.

As shown in Figure 5, the Off-Policy Hint method underperforms both HiPO and the GRPO baseline, a failure explained by the collapse of tool-use turns in Figure 4(b). This collapse occurs because static hints derived from the untrained base model are inherently simplistic. Enforcing these "weak" priors suppresses exploration, trapping the policy in local optima where it mimics trivial solutions rather than developing the complex reasoning chains required for harder problems. In contrast, HiPO establishes a self-reinforcing curriculum where hints evolve dynamically with the policy. As evidenced by rising tool usage, the "teacher" improves alongside the "student," enabling autonomous mastery without reliance on stronger external models.

## 5.6 SAMPLE EFFICIENCY ANALYSIS

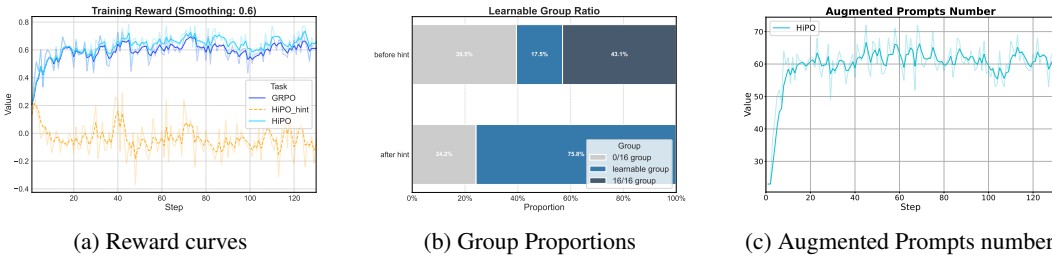

(a) Reward curves      (b) Group Proportions      (c) Augmented Prompts number

Figure 6: Visualization of HiPO's training dynamics. (a) Training reward curves for HiPO and GRPO. The HiPO_hint represents the reward after group substitution. (b) The proportion of learnable groups versus ineffective groups within a batch, before (left) and after (right) applying HiPO. (c) The curve of the number of prompts enhanced by the HiPO method during training.

HiPO is designed to enhance sample efficiency by resolving the "signal collapse" problem endemic to group-based methods. Our training analysis in Figure 6 empirically confirms this. GRPO wastes computation on "unlearnable groups", where a staggering 82.5% of all groups (the sum of 0/16 and 16/16 groups) yield a null gradient. In contrast, our hint mechanism transforms these into valuable learning opportunities. As shown in Figure 6b, HiPO elevates the proportion of learnable groups to 75.8%. This self-sustaining process remains stable throughout training, evidenced by a consistently high number of augmented prompts (Figure 6c), which ensures a dense and reliable learning signal.

The impact of this efficiency is evident in the training reward curves shown in Figure 6a. HiPO's policy (blue) consistently outperforms the GRPO baseline, demonstrating superior learning. Critically, this performance gain is achieved on a stable and challenging curriculum. The HiPO hint curve (orange), representing reward from hint-guided rollouts, remains consistently low. This signals that HiPO continuously drives the model to learn from difficult, partially-completed trajectories rather than saturating on easy ones. Consequently, overall capability rises (blue curve) while the learning signal stays challenging, ensuring robust and sustained improvement.

## 6 CONCLUSION

We addressed the critical challenge of exploration stagnation in RLVR by introducing HiPO, a framework built on the paradigm of endogenous Self-Hint. HiPO transforms rare, stochastically-found successes into an on-policy curriculum, converting a sparse reward landscape into a dense, contrastive learning signal. This self-teaching mechanism not only significantly outperformed strong baselines across challenging reasoning benchmarks but also demonstrably fostered higher policy entropy and more complex reasoning chains. Our work establishes that a model can effectively bootstrap its own learning from endogenous successes, reducing the reliance on external expert data.

The principles of HiPO pave the way for a more scalable and autonomous paradigm of skill acquisition. By demonstrating that models can become active participants in their own education, our work provides a robust foundation for training self-improving agents capable of mastering intricate tasks in any domain where success is a rare and hard-won event. This highlights a powerful path toward more capable and independent AI.

ACKNOWLEDGMENTS

This work was supported in part by the National Natural Science Foundation of China (U23B2055, 62276077, 62350710797), in part by the Guangdong Basic and Applied Basic Research Foundation under Grant 2024A1515011205, andin part by Shenzhen Science and Technology Program (KQTD20240729102154066, ZDSYS20230626091203008).

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

## A    LLM Usage

We used large language models (LLMs) only to polish grammar and improve the clarity of the manuscript. All research ideas, experiments, and analyses were conducted by the authors.

## B    Comparison with DAPO

We compare HiPO against DAPO (Yu et al., 2025), a strong baseline employing dynamic sampling to tackle sparse rewards. To accurately reflect computational cost, Figure 7 evaluates performance against prompt volume.

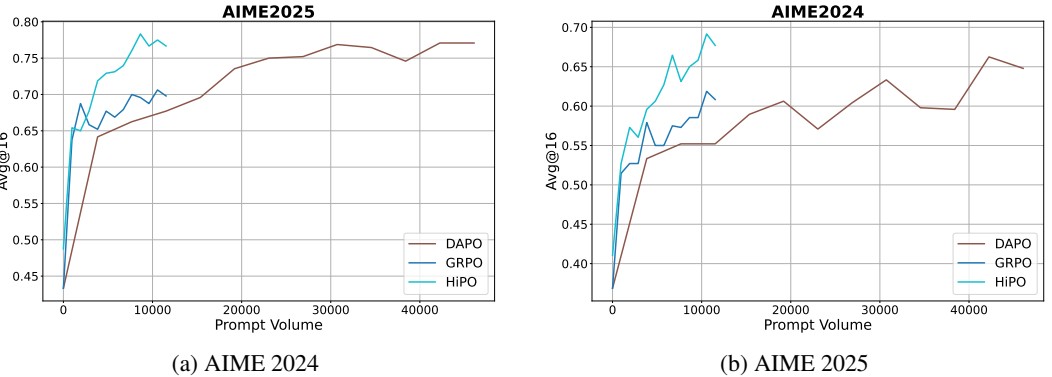

(a) AIME 2024       (b) AIME 2025

Figure 7: Performance curves (Avg@16) on AIME 2024 (a) and AIME 2025 (b) plotted against the total Prompt Volume. The trajectories correspond to GRPO (blue), HiPO (cyan), and DAPO (brown).

As shown in Figure 7, HiPO achieves superior convergence with a fraction of the data consumption. The extended tail of DAPO reveals the high cost of its dynamic sampling mechanism: to secure positive rewards, it must perform extensive rejection sampling, consuming approximately $4\times$ the trajectory budget per update step. While DAPO relies on brute-force enumeration to locate sparse successes, HiPO employs signal amplification. By repurposing a single stochastic success into a group-wide hint, HiPO extracts dense gradients from standard batches, eliminating the need for wasteful resampling and establishing a significantly more efficient learning paradigm.

## C    Robustness to Ultra-Sparse Rewards

A primary concern regarding RLVR in complex domains is the "cold start" problem, where ultra-sparse rewards might starve the model of learning signals. To empirically evaluate HiPO's robustness in such regimes, we conducted a stress test using the top 10% most difficult problems from the Omni-Math dataset (Gao et al., 2024) to train a Qwen3-8B base model. This setup simulates a near-zero success rate environment where standard exploration often stagnates.

The results in Figure 8 demonstrate that HiPO effectively mitigates signal scarcity through batch-level signal broadcasting. As illustrated in Figure 8(a), the initial generation phase (blue bars) is dominated by groups with zero successes, which would typically yield a null gradient. However, HiPO identifies rare stochastic successes present within the global batch and repurposes them to generate hinted groups. The cyan bars show a drastic reduction in these unlearnable groups after intervention, indicating that the method effectively amplifies sparse signals by substituting failed trajectories with actionable, hint-guided ones. This mechanism ensures that even when success is statistically rare, the optimization process remains supplied with valid gradients. Figures 8(b) and (c) confirm that this signal rectification translates into tangible learning progress, with the model maintaining a steady upward trend on both AIME 2024 and AIME 2025 benchmarks despite the extreme difficulty of the training distribution.

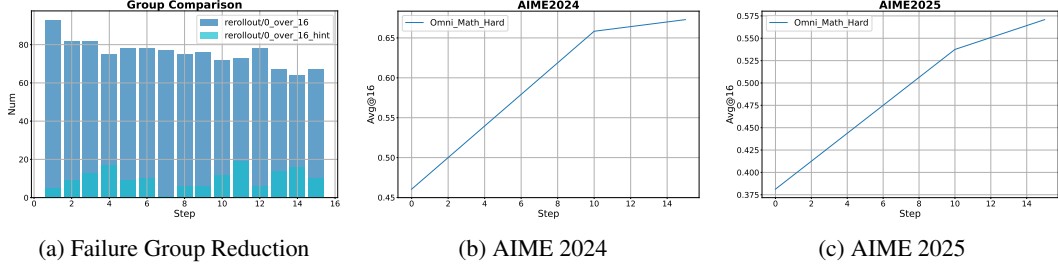

(a) Failure Group Reduction      (b) AIME 2024      (c) AIME 2025

Figure 8: HiPO in Ultra-Sparse Regimes (Omni-Math-Hard). (a) The number of "Total Failure" groups (0/16 success) in a batch before (Blue) and after (Cyan) HiPO intervention. HiPO drastically reduces the proportion of null-signal groups. (b-c) Despite the extreme difficulty, the model maintains a steady upward learning trend on AIME benchmarks, confirming that the feedback loop remains unbroken.

## D    RANDOM HINT SETTING

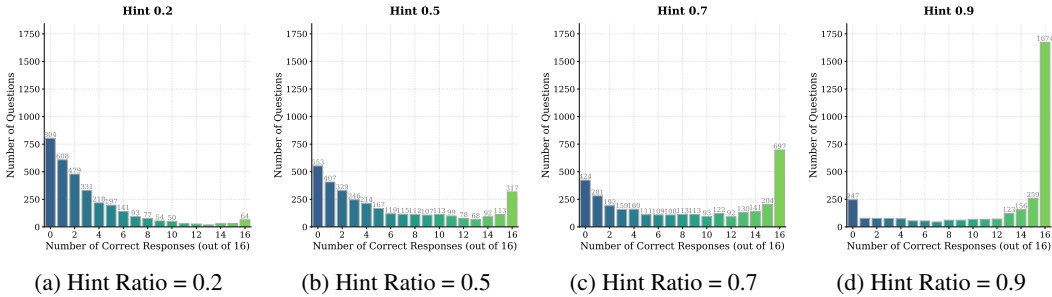

(a) Hint Ratio = 0.2      (b) Hint Ratio = 0.5      (c) Hint Ratio = 0.7      (d) Hint Ratio = 0.9

Figure 9: Distribution of the number of correct responses over 16 attempts ($n/16$) for different fixed hint ratios. These plots illustrate how increasing the hint ratio shifts the distribution towards perfect scores, inadvertently creating less effective learning signals.

As empirically demonstrated in Figure 9, hint ratios exceeding 0.5 cause a sharp increase in the frequency of perfect-score (16/16) groups. This outcome is detrimental because it induces "signal collapse." In a group where all trajectories succeed, the reward variance is zero, which nullifies the advantage estimate ($\hat{A}_{j,i} = 0$) for all samples and causes the policy gradient to vanish. Furthermore, excessively long hints reduce the problem to a trivial completion task, preventing the model from learning the complex intermediate reasoning steps and suppressing meaningful exploration of the solution space.

Using a single, fixed hint ratio creates a critical credit assignment pathology, which can be shown formally. Consider a hinted group $T_j = \{\tau_{j,1}, \ldots, \tau_{j,n}\}$ where all trajectories share an identical prefix $H = (o_0, \ldots, o_{k-1})$ because they are derived from a single source trajectory with a fixed ratio. In GRPO, the total policy gradient for any token $o_t$ within this shared prefix ($t < k$) is the sum of its gradients from each trajectory in the group. Since the advantage $\hat{A}_{j,i}$ is constant for all tokens within a trajectory $\tau_{j,i}$, and the conditional probability $\pi_\theta(o_t|s_t)$ is identical for all trajectories at this step, the total gradient for token $o_t$ is:

$$\nabla_\theta J(o_t) = \sum_{i=1}^{n} \hat{A}_{j,i} \nabla_\theta \log \pi_\theta(o_t|s_t) = \left( \sum_{i=1}^{n} \hat{A}_{j,i} \right) \nabla_\theta \log \pi_\theta(o_t|s_t). \tag{10}$$

The core issue lies in the summed advantage term. By definition, the advantage $\hat{A}_{j,i} = (R(\tau_{j,i}) - \mu_{\tau_j})/(\sigma_{\tau_j} + \epsilon)$, where $\mu_{\tau_j}$ is the mean reward of the group. The sum of all advantages within the group is therefore:

$$\sum_{i=1}^{n} \hat{A}_{j,i} = \frac{1}{\sigma_{\tau_j} + \epsilon} \sum_{i=1}^{n} (R(\tau_{j,i}) - \mu_{\tau_j}) = \frac{1}{\sigma_{\tau_j} + \epsilon} \left( \left( \sum_{i=1}^{n} R(\tau_{j,i}) \right) - n \cdot \mu_{\tau_j} \right) = 0. \tag{11}$$

This demonstrates that the aggregate advantage signal for any token in the shared prefix is precisely zero, resulting in a null gradient. The model is thus unable to learn that the shared prefix is a valuable reasoning path. Randomly sampling the hint ratio breaks this pathological symmetry by ensuring the prefixes are not identical, which allows for a meaningful, non-zero gradient to be assigned.

## E    GENERALIZATION TO OUT-OF-DISTRIBUTION TASKS

We evaluated HiPO on HumanEval to assess its generalization capabilities beyond the mathematical domain. Despite the training data consisting exclusively of mathematical problems—making code generation a fully out-of-distribution (OOD) task, HiPO achieves a Pass@1 score of 71.3%, outperforming the Qwen3-8B base model (70.7%). This demonstrates that the enhanced reasoning patterns fostered by HiPO are robust and transferable, yielding performance improvements even on tasks unseen during training.

Table 2: Pass@1 (%) on the HumanEval benchmark. HiPO demonstrates positive generalization on this OOD coding task, outperforming the base model.

| Model | Pass@1 |
|---|---|
| Qwen3-8B (Base) | 70.7 |
| GRPO | 71.3 |
| HiPO (Ours) | **71.3** |

## F    CASE STUDY

To provide a concrete illustration of the behavioral differences between HiPO and the GRPO baseline, we present a qualitative case study. The following Table 3, Table 4 and Table 5 contrasts the reasoning trajectories of both models on a challenging mathematical problem that requires both insight and precise execution. This side-by-side comparison highlights HiPO's ability to perform strategic exploration to simplify the problem's core, versus GRPO's tendency to get locked into a linear, computationally intensive, and ultimately fragile reasoning path.

As the comparison demonstrates, HiPO's ability to perform strategic exploration leads to a more robust and efficient solution path. In contrast, the baseline model's linear, brute-force approach results in a fragile reasoning chain that ultimately collapses due to its own complexity, perfectly illustrating the "exploration stagnation" that our method is designed to overcome.

Table 3: A second comparative case study. This table illustrates HiPO's strategy of leveraging external tools (a code interpreter) to definitively verify its algebraic insight. In contrast, the GRPO baseline performs a robust but purely algebraic verification, re-solving the problem with a different, more complex method to ensure correctness.

| [Question] | |
|---|---|
| Positive real numbers $x$ and $y$ satisfy $y^3 = x^2$ and $(y-x)^2 = 4y^2$. What is $x+y$? | |
| **HiPO Trajectory (Our Method)** | **GRPO Trajectory (Baseline)** |
| *Response:* | *Response:* |
| ...Wait, $(y-x)^2 = 4y^2 \to$ taking square roots gives $\|y - x\| = 2y$. | ...Let's take the square root of both sides... we get $\|y - x\| = 2y$. |
| ...Case 1: $y - x = 2y \to x = -y$. But x and y are positive, so this can't be possible... | ...Case 1: $y - x = 2y \to x = -y$. But x and y are positive, so this is impossible. |
| Case 2: $y - x = -2y \to x = 3y$. | Case 2: $y - x = -2y \to x = 3y$. So this is the valid case. |
| So from this, x equals 3y. Now, substitute this into equation 1. | Substitute $x = 3y$ into... $y^3 = (3y)^2 \to y^3 = 9y^2$. ...so $y = 9$. |
| $y^3 = (3y)^2 \to y^3 = 9y^2 \to y = 9$. | Then $x = 3y = 3 \times 9 = 27$. Therefore, $x + y = 27 + 9 = 36$. |
| So y=9. Then x=3y=27. Therefore, $x + y = 27 + 9 = 36$. | **Wait, but let me check again if there's another way to approach this...** |
| **But perhaps the code can solve this system of equations.** | Alternatively, maybe expanding equation 2. |
| | $(y - x)^2 = y^2 - 2xy + x^2 = 4y^2$. |
| ```python
import sympy as sp
x, y = sp.symbols('x y', positive=
    True)
eq1 = sp.Eq(y**3, x**2)
eq2 = sp.Eq((y - x)**2, 4*y**2)
solution = sp.solve((eq1, eq2), (x,
    y))
result = solution[0][0] + solution
    [0][1]
# result is 36
``` | ... $x^2 - 2xy - 3y^2 = 0$. Let me treat this as a quadratic equation in x... Using quadratic formula: $x = [2y \pm \sqrt{(2y)^2 - 4(1)(-3y^2)}]/2$ $x = [2y \pm \sqrt{16y^2}]/2 = [2y \pm 4y]/2$. ...only $x = 3y$ is valid. So same result as before. So that's correct. |
| **Yes, the code gives 36.** So that confirms the answer. The final answer is $\boxed{36}$. | All approaches lead to x=27, y=9, sum 36. The final answer is $\boxed{36}$. |

Table 4: A case study in problem restructuring. The HiPO trajectory shows the use of a key algebraic insight to simplify the problem's structure, leading to a straightforward solution. In contrast, the baseline's linear approach gets entangled in more complex and error-prone calculations.

---

**[Question]**

Find the number of triples of nonnegative integers $(a, b, c)$ satisfying $a + b + c = 300$ and

$$a^2b + a^2c + b^2a + b^2c + c^2a + c^2b = 6,000,000.$$

---

| HiPO Trajectory (Our Method) | GRPO Trajectory (Baseline) |
|---|---|
| *Response:* | *Response:* |
| ...This leads to the relation 'abc - 100(ab + bc + ca) = -2,000,000'. | ...This leads to the relation 'abc - 100(ab + bc + ca) = -2,000,000'. |
| **This seems a bit abstract. How can I proceed from here?** | **But how can I proceed from here? Let me think.** |
| *Alternatively, maybe think of the left-hand side as (a-100)(b-100)(c-100)?* ... | Let me check the case where one variable is zero first... This gives 6 solutions. |
| $$(a - 100)(b - 100)(c - 100) = 0.$$ | Now, let me think about other cases... |
| **Oh! That's a big simplification!** ...This is a counting problem I can solve with inclusion-exclusion. | $$P'(c - 100) = -100(c^2 - 300c + 20000).$$ |
| Number of solutions is: $3 \times 201 - 3 \times 1 + 1 = 601$. | Let me factor the right-hand side... |
| But to make sure, perhaps code it. | $$P'(c - 100) = -100(c - 200)(c - 100).$$ |
| | ...The discriminant is $D = (S' - 200)^2$, which is a perfect square. Exactly! |

HiPO column code:

```
def count_solutions():
    # Solutions where a=100: b+c
    =200 -> 201
    count = 3 * 201
    # Subtract overlaps where a=b
    =100
    count -= 3
    # Add back the case where a=b=c
    =100
    count += 1
    return count
print(count_solutions())
```

GRPO column continued:

...So for each $c$ from 1 to 199, there are two ordered triples... except $c = 100$. So total solutions is $1 + 198 \times 2 = 397$.

**Wait, this is a critical point... This is a big mistake in the earlier reasoning...**

The total number of solutions is $6 + 397 = 403$. The final answer is $\boxed{403}$.

HiPO column continued:

**Interpreter Output:** 601

The code confirms my reasoning. The final answer is $\boxed{601}$.

Table 5: A case study on strategy selection. HiPO correctly identifies the problem as best suited for a computational brute-force search, leading to the correct answer. The baseline attempts a more complex analytical shortcut, but a subtle reasoning error leads to an incorrect result, highlighting the robustness of HiPO's strategy.

---

**[Question]**

Let $b \geq 2$ be an integer. Call a positive integer $n$ *b-eautiful* if it has exactly two digits when expressed in base $b$ and these two digits sum to $\sqrt{n}$. For example, 81 is 13-*eautiful* because $81 = \underline{6}\,\underline{3}_{13}$ and $6 + 3 = \sqrt{81}$. Find the least integer $b \geq 2$ for which there are more than ten *b-eautiful* integers.

| **HiPO Trajectory (Our Method)** | **GRPO Trajectory (Baseline)** |
|---|---|
| *Response:* | *Response:* |
| Okay, I need to solve this problem... For a number to have exactly two digits in base b, it must satisfy $b \leq n < b^2$. Let the digits be d1 and d2. Then $n = d1 \cdot b + d2$ and $d1 + d2 = \sqrt{n}$. **So the plan is: For each base b starting from 2 upwards... iterate over all possible d1 and d2... check if (d1+d2)² == n... and count them.** This seems suitable for code. | Okay, let's tackle this problem... Let n = a*b + c. The condition is $\sqrt{ab + c} = a + c$. Let s = a+c. Then $s^2 = ab + (s - a)$, which gives $a = \frac{s(s-1)}{b-1}$. For a to be an integer, b-1 must divide s(s-1). Since s and s-1 are coprime, **b-1 must divide s or s-1**. ...This seems to imply there are very few solutions for any b. |

```
def find_min_base():
    b = 2
    while True:
        count = 0
        for d1 in range(1, b):
            for d2 in range(0, b):
                n = d1 * b + d2
                sum_d = d1 + d2
                if sum_d * sum_d == n:
                    count += 1
        if count > 10:
            return b
        b += 1
result = find_min_base()
print(result)
```

**Wait, let me check the example. b=13, n=81.** Digits are 6,3. s=9. b-1=12. Here, $s(s - 1) = 9 \times 8 = 72$. 12 divides 72, but 12 does not divide 9 or 8. **Ah! My reasoning was incorrect.** So I need to check for each s if b-1 divides s(s-1). Let me try to code this.

```
# Tries to write code
# ... but it has syntax errors
def count_beautiful(b):
    #...
# ... code fails to run
```

**Interpreter Output:** 211
**Wait, the code returns 211? That seems high.**
Let me check for b=3.
d1=1, d2=1: n=4, sum=2. 2²=4. Yes.
...other pairs fail. So for b=3, count is 1. The code logic seems to be working. So the answer is $\boxed{211}$.

**At this point, it's clear that without writing the code, it's hard to proceed.** Let me try to estimate for b=61... This is getting too complex. The answer is $\boxed{121}$.

