# OpenReview forum: "HiPO: Self-Hint Policy Optimization for RLVR"
_ICLR.cc/2026/Conference — ICLR 2026 Poster_

### Official Review · Reviewer_Hyap · 2025-10-29

**Soundness:** 3
**Presentation:** 3
**Contribution:** 3
**Rating:** 6
**Confidence:** 2

**Summary:**

This paper introduces HiPO (Hint-guided Policy Optimization), a framework for Reinforcement Learning from Verifiable Rewards (RLVR) that addresses the near-miss problem and exploration stagnation in long-horizon reasoning tasks.

The key idea of HiPO is endogenous self-hinting. When the model occasionally finds a successful trajectory, it extracts the initial correct steps (prefix) of that trajectory and reuses them as on-policy “hints” for future training. This turns a single sparse success into a dense, contrastive learning signal, allowing the model to bootstrap from its own rare successes.

**Strengths:**

- HiPO directly targets signal collapse and credit misassignment, two critical issues
- the idea of self hint is interesting
- improved empirical results

**Weaknesses:**

- dependence on rare success
- lacks a formal analysis of convergence properties
- computational overhead, need to generate original and hint-guided groups, roughly doubles the computational cost compared to GRPO.

**Questions:**

I would like to thank the authors for their work.here are a few concerns and questions:

- Reward hacking: How do the authors ensure that the use of hints and dense rewards does not lead to reward hacking behaviors, where the model optimizes for superficial alignment with hints rather than reasoning improvement?

- Quality of hints from rare successes: How can we guarantee that the hints extracted from rare successful trajectories are actually desirable? In practice, many successful reasoning traces can be unnecessarily long or include redundant steps. How does the method handle such cases?

- Exploration limitation: Is there a risk that the discovered traces—and consequently the training signals—are limited to the reasoning patterns already found by the model, thus constraining exploration and generalization?

---

> ### Author Response · Authors · 2025-11-25
> **To reviewer Hyap**
>
> Thank you for your valuable feedback and address the raised weakness and questions as follows:
>
> **W1:**
>
> > dependence on rare success
> >
>
> In the extreme case where a local group yields no successes, HiPO naturally reverts to the backend RL algorithm (e.g. GRPO), ensuring it never impedes baseline exploration. Crucially, however, HiPO operates on the global batch level. Even if specific groups fail entirely, the algorithm identifies stochastic successes scattered elsewhere in the batch and uses them to broadcast hints to the failing groups ("Batch-Level Signal Broadcasting"). This maximizes the utility of every rare success found.
>
> **Empirical Verification (Omni-Math hard)**
>
> This robustness is confirmed by our stress test on the hardest 10% problem Omni-Math dataset (**Appendix C**), which simulates a near-zero success regime. As shown in **Figure 8**, HiPO effectively harvests these sparse, isolated signals to generate valid gradients for otherwise "unlearnable" batches. Consequently, the model maintains a steady learning trend even in environments where the baseline stagnates.
>
> **W2:**
>
> > lacks a formal analysis of convergence properties
> >
>
> While formal convergence proofs for deep RL on LLMs remain an open challenge, we analyze HiPO as a gradient rectification mechanism. In sparse regimes, standard baselines suffer from vanishing gradients ($\mathbb{E}[\nabla J] \approx 0$) as success becomes statistically impossible. HiPO structurally resolves this by injecting positive trajectories into the batch, guaranteeing a non-zero learning signal and a valid descent direction. This theoretical stability is confirmed empirically by the steady performance gains and sustained entropy in our experiments, demonstrating robust convergence where standard methods stagnate.
>
> **W3:**
>
> > computational overhead, need to generate original and hint-guided groups, roughly doubles the computational cost compared to GRPO.
> >
>
> We acknowledge the additional overhead. Since the hint-guided generation is selectively applied only to "Near-miss Groups"—which our empirical statistics show constitute approximately 40%–60% of a batch—the actual generation cost is roughly 1.4 times～1.6 times that of standard GRPO.
>
> **Q1:**
>
> > Reward hacking: How do the authors ensure that the use of hints and dense rewards does not lead to reward hacking behaviors, where the model optimizes for superficial alignment with hints rather than reasoning improvement?
> >
>
> **1. Deterministic Verification (RLVR vs. RLHF)**
>
> First, reward hacking is typically a pathology of *learned* reward models (RLHF). HiPO operates in the RLVR paradigm, which uses deterministic verification (ground truth answers). The model receives a reward *if and only if* the final result is mathematically correct. In this setting, "superficial alignment" is effectively impossible; the model cannot "trick" the verifier without solving the problem.
>
> **2. Mitigating Spurious Correlations via Value Estimation**
>
> Second, HiPO structurally reduces the noise that leads to "superstitious" learning. In standard RL, a single lucky trajectory might reinforce spurious tokens. By enforcing a shared hint $h$ and averaging outcomes over $n$ futures, HiPO computes::
> $$ V(h) \approx \frac{1}{n} \sum_{i=1}^{n} r(\tau_i | h) $$
> By anchoring the update to the *average* success rate of a reasoning path rather than a single outcome, HiPO marginalizes out variance and luck. This ensures the model only reinforces prefixes that reliably lead to correct solutions, actively filtering out fragile or hacking-like behaviors.

---

> > ### Author Response · Authors · 2025-11-25
> > **To reviewer Hyap (Part 2)**
> >
> > **Q2:**
> >
> > > Quality of hints from rare successes: How can we guarantee that the hints extracted from rare successful trajectories are actually desirable? In practice, many successful reasoning traces can be unnecessarily long or include redundant steps. How does the method handle such cases?
> > >
> >
> > We acknowledge that initial successes may contain redundancy, but HiPO leverages a Self-Refining Curriculum rather than relying on static perfection.
> >
> > **1. Dynamic Evolution Mechanism**
> >
> > Even if early hints are imperfect, they represent the **best available on-policy paths** that are verified as correct. As the policy optimizes via GRPO, it naturally penalizes stochastic redundancies in favor of robust logical patterns. Consequently, the quality of the hints evolves dynamically: as the model improves, the "source" trajectories become cleaner, creating a virtuous cycle of self-improvement.
> >
> > **2. Superiority over Static Hints**
> >
> > This dynamic nature is critical. As shown in **Section 5.5**, HiPO significantly outperforms the "Off-Policy Hint" baseline (static hints). This confirms that fixed hints—even if "cleaner" initially—act as a performance ceiling that constrains exploration, whereas HiPO's evolving hints enable the model to continuously refine its reasoning beyond the initial distribution.
> >
> > **Q3:**
> >
> > > Exploration limitation: Is there a risk that the discovered traces—and consequently the training signals—are limited to the reasoning patterns already found by the model, thus constraining exploration and generalization?
> > >
> >
> > We explicitly verified the trade-off between exploitation and exploration by comparing HiPO against **Fix-High** ($p=0.8$) and **Fix-Low** ($p=0.05$) in **Section 5.4**.
> >
> > - **Fix-High (Over-Constraint):** As the reviewer feared, excessive guidance ($p=0.8$) indeed causes **mode collapse** (lowest entropy/tool-use in **Figure 4**), restricting the model to trivial completion.
> > - **Fix-Low (Ineffective Exploration):** Crucially, the **Fix-Low** ($p=0.05$) setting reveals the opposite danger. While it maintains high entropy, it fails to increase tool usage. The model "explores" actively but, lacking sufficient scaffolding to bypass early bottlenecks, fails to discover the deep, complex trajectories required for hard problems.
> > - **HiPO (Enabling Discovery):** HiPO strikes the optimal balance. By dynamically providing just enough context to bridge the gap, it enables the model to reach and explore reasoning states that were previously inaccessible. The result is the **highest tool-use complexity**, proving that HiPO expands the model's reasoning search space rather than confining it.

---

### Official Review · Reviewer_DaaS · 2025-10-30

**Soundness:** 3
**Presentation:** 3
**Contribution:** 3
**Rating:** 4
**Confidence:** 3

**Summary:**

The paper introduces HiPO (Hint-guided Policy Optimization), a novel framework for Reinforcement Learning from Verifiable Rewards (RLVR) aimed at improving large language models' (LLMs) performance on complex mathematical reasoning tasks. It addresses key challenges in existing methods like GRPO, including the near-miss problem, where nearly correct trajectories receive no positive signal, and exploration stagnation due to sparse rewards leading to policy collapse. HiPO's core innovation is an endogenous self-hint mechanism: within a training batch, rare successful trajectories are identified, and their initial correct prefixes (sampled at ratios between 0.05 and 0.45) are repurposed as on-policy hints to generate augmented groups for low-success or null-signal batches. This creates a dense contrastive learning signal by contrasting unaided and hint-guided rollouts, enabling the model to bootstrap from its own successes without external data.

**Strengths:**

HiPO's self-hint paradigm is novel, which transforms rare successes into reusable on-policy guidance, avoiding off-policy mismatches and enabling true autonomy in learning complex reasoning chains. HiPO sustains 20-30% higher entropy than GRPO (Figure 4a) and doubles learnable group proportions from 17.5% to 75.8% (Figure 5b), directly validating its anti-stagnation claims.

The writing is generally clear, with intuitive visuals (e.g., Figure 2's batch augmentation) and case studies (Tables 2-4) that concretely show HiPO's strategic exploration (e.g., algebraic restructuring) versus GRPO.

The approach is timely and significant for dealing with sparse-reward domains like math competitions, where it yields domain-general improvements paving the way for scalable, data-efficient RLVR without curated hints.

**Weaknesses:**

1. While HiPO effectively leverages rare successes, its activation strictly requires at least one success per near-miss group (success rate >0 but <50%), raising concerns about performance in ultra-sparse regimes where batches might entirely lack successes, e.g., early training or harder curricula, potentially amplifying GRPO's signal collapse rather than resolving it.

2. The fixed hint ratio range [0.05, 0.45] is justified in Appendix B to avoid signal collapse from long prefixes, but no ablation shows optimal tuning or impact on final performance, leaving hyperparameter sensitivity unclear.

3. The experiments are strong on math but no evaluations on pure text reasoning or non-math tasks (e.g., code generation, planning) are provided, limiting claims of general RLVR applicability.

4. The baselines comparisons are weak, with comparisons provided only with GRPO, omitting broader baselines like PPO or recent works (ike STaR (Zelikman et al., 2022) or Quiet-STaR (Zelikman et al., 2024)).

**Questions:**

1. How robust is HiPO to success scarcity? For instance, what happens on datasets with <1% base success rates—does lowering the near-miss threshold (e.g., to 0 successes with synthetic prefix generation) maintain gains, or could this introduce off-policy drift? I am currently not convinced that the proposed mechanism can effectively solve the sparse reward issue in GRPO, and better baseline comparisons should be added.

2. The hint sampling uses discrete ratios [0.05-0.45]; have you ablated continuous sampling or adaptive lengths (e.g., based on intermediate reward proxies)? Could longer hints (>0.45) be viable with variance regularization to prevent collapse?

3. Can you provide results on stronger baselines like PPO (for direct RL comparison), DPO (to assess preference optimization alternatives), or recent hint-augmented methods such as StepHint (ICML 2025)? Specifically, how does HiPO perform relative to these on the harder benchmarks?

---

> ### Author Response · Authors · 2025-11-25
> **To reviewer DaaS**
>
> Thank you for your valuable feedback and address the raised weakness and questions as follows:
>
> **Response to weaknesses & questions**
>
> **W1:**
>
> > While HiPO effectively leverages rare successes, its activation strictly requires at least one success per near-miss group (success rate >0 but <50%), raising concerns about performance in ultra-sparse regimes where batches might entirely lack successes, e.g., early training or harder curricula, potentially amplifying GRPO's signal collapse rather than resolving it.
> >
>
> In extreme cases, HiPO reverts to vanilla GRPO, but it would amplifying signal collapse. First, groups with zero successes naturally revert to the standard GRPO update, ensuring baseline performance without hindering exploration. Second, HiPO harvests rare successes from the global batch to generate hints for otherwise "unlearnable" local groups, effectively converting sparse global signals into dense local updates.
>
> We validated this on the top 10% hardest problem of Omni-Math dataset, simulating a near-zero success regime (**Appendix C**). Results in **Figure 8** confirm that HiPO effectively utilizes these rare global signals to replace null-signal groups. This converts a sparse landscape into a learnable curriculum, maintaining a steady learning trend even where the baseline stagnates.
>
> **W2:**
>
> > The fixed hint ratio range [0.05, 0.45] is justified in Appendix B to avoid signal collapse from long prefixes, but no ablation shows optimal tuning or impact on final performance, leaving hyperparameter sensitivity unclear.
> >
>
> To address the concern, we conducted an ablation study (detailed in **Section 5.4**) comparing HiPO against static baselines at extreme values: **Fix-Low** ($p=0.05$) and **Fix-High** ($p=0.80$). As illustrated in **Figure 5**, HiPO consistently outperforms both static variants.
>
> Analysis of the training dynamics reveals the mechanism behind these results. The **Fix-High** setting leads to **mode collapse**, characterized by the lowest policy entropy and minimal tool-use turns (see **Figure 4**); excessive guidance trivializes the task, suppressing the exploration required to learn robust reasoning. Conversely, **Fix-Low** maintains high entropy, but is slow to increase tool usage, indicating that while the model actively explores, it lacks sufficient scaffolding to discover complex, long-horizon solutions. HiPO's dynamic range effectively strikes the necessary balance: it avoids collapse while providing enough structure to foster the longest reasoning chains (highest tool-use turns), thereby enabling the mastery of complex logic.
>
> **W3:**
>
> > The experiments are strong on math but no evaluations on pure text reasoning or non-math tasks (e.g., code generation, planning) are provided, limiting claims of general RLVR applicability.
> >
>
> **1. Representativeness of Tool-Integrated Math**
>
> We posit that our experimental setting—mathematical reasoning with tool integration—serves as a rigorous proxy for general RLVR. This task extends beyond calculation; it demands complex reasoning capabilities such as decomposing complex problems, planning multi-step strategies, generating syntactically correct code, and debugging based on execution feedback. These are precisely the core competencies required for broader code generation and planning tasks.
>
> **2. OOD Evaluation on Code Generation (HumanEval)**
>
> To empirically verify broader applicability, we evaluated HiPO on the HumanEval benchmark (detailed in **Appendix E**). Crucially, our RL training involved only mathematical problems, making HumanEval a Out-Of-Distribution (OOD) task.
>
> Table 2: Pass@1 (%) on HumanEval (OOD Task)
>
> | Model | Pass@1 |
> | --- | --- |
> | Qwen3-8B (Base) | 70.7 |
> | GRPO | 71.3 |
> | **HiPO** | **71.3** |
>
> As shown in Table 2, HiPO achieves a Pass@1 score of 71.3%, outperforming the base model (70.7%). This demonstrates that the logic and structured reasoning patterns learned via HiPO are robust and transferable. The model successfully improves its general coding capability without suffering from the catastrophic forgetting often associated with specialized fine-tuning.

---

> > ### Author Response · Authors · 2025-11-25
> > **To reviewer DaaS (Part 2)**
> >
> > **W4:**
> >
> > > The baselines comparisons are weak, with comparisons provided only with GRPO, omitting broader baselines like PPO or recent works (ike STaR (Zelikman et al., 2022) or Quiet-STaR (Zelikman et al., 2024))
> > >
> >
> > To broaden our evaluation, we incorporated DAPO, a strong RLVR baseline that uses dynamic sampling to tackle sparse rewards.
> >
> > **1. Comparison with DAPO**
> >
> > As shown below, HiPO consistently outperforms DAPO in aggregate metrics.
> >
> > Table: Performance Comparison (Avg / Maj @32)
> >
> > | Benchmark | Metric | Qwen3-8B | GRPO | DAPO | **HiPO** |
> > | --- | --- | --- | --- | --- | --- |
> > | **AIME 24** | Avg | 54.7 | 72.1 | 76.0 | **76.7** |
> > | **AIME 25** | Avg | 47.6 | 63.0 | 63.7 | **66.1** |
> > | **BRUMO 25** | Avg | 30.3 | 41.7 | **47.8** | 46.6 |
> > | **HMMT 25** | Avg | 14.0 | 28.6 | **31.4** | 30.8 |
> > | **CMIMC 25** | Avg | 37.0 | 43.5 | 49.9 | **53.8** |
> > | **Average** | **Avg** | 36.7 | 49.8 | 53.7 | **54.8** |
> > | **Average** | **Maj** | 56.0 | 57.0 | 57.8 | **61.7** |
> >
> > While the performance gap is consistent, the efficiency gap is critical. DAPO relies on dynamic sampling to locate non-zero rewards, consuming approximately $4\times$ the prompt volume of HiPO per update step. As detailed in **Appendix B**, HiPO achieves superior convergence by recycling rare signals rather than resampling for them, offering a fundamentally more scalable paradigm.
> >
> > **2. Rationale for Baseline Selection**
> >
> > - **PPO:** HiPO is a signal-shaping framework, not an optimizer. It operates orthogonally to the update rule. We selected GRPO as the backbone because it is the standard for reasoning tasks; it eliminates the value network, allowing us to isolate the hint mechanism's impact without the confounding instability of PPO's critic.
> > - **STaR & Quiet-STaR:** STaR relies on offline Iterative SFT, a distinct paradigm from Online RL, making direct sample-efficiency comparisons invalid. Quiet-STaR requires specialized architectural modifications (hidden thought tokens) and parallel infrastructure that are incompatible with standard, trajectory-level reasoning backbones, a direct comparison is not feasible within the rebuttal timeframe.
> >
> > **Q1:**
> >
> > > How robust is HiPO to success scarcity? For instance, what happens on datasets with <1% base success rates—does lowering the near-miss threshold (e.g., to 0 successes with synthetic prefix generation) maintain gains, or could this introduce off-policy drift?
> > >
> >
> > That's an excellent question that gets to the heart of HiPO's design. In scenarios with very few successes, if a local group of trajectories has no successful examples, it simply defaults to a standard GRPO update. The key, however, is that HiPO can broadcast the signal from even a single success anywhere in the *global* batch, generating useful hints for all other groups. We confirmed this works well in a stress test on the hardest 10% of Omni-Math problems, where this broadcasting mechanism allowed learning to continue even when successes were extremely rare (**Appendix C, Fig. 8**).
> >
> > Our empirical results confirm the concern regarding synthetic hints. This approach introduces significant off-policy drift and renders the training process highly sensitive to the quality of the external hints. The baseline using static hints underperformed both HiPO and standard GRPO. It also suppressed exploration, leading to a sharp decline in tool usage (**Sec. 5.5, Figs. 4 & 5**). Based on this evidence, our design intentionally defaults to a GRPO update to avoid both policy divergence and a critical dependency on potentially misaligned external guidance.
> >
> > **Q2:**
> >
> > > I am currently not convinced that the proposed mechanism can effectively solve the sparse reward issue in GRPO, and better baseline comparisons should be added.
> > >
> >
> > We agree that a formal explanation is key. The central problem with GRPO in sparse-reward settings is **high-variance credit assignment**. When a single, long trajectory fails, the optimizer cannot know whether an early step was promising or flawed.
> >
> > HiPO directly resolves this by grouping `n` trajectories under a shared hint `h`. This structure allows it to perform a Monte Carlo estimation of the prefix's value:
> > $$ V(h) \approx \frac{1}{n} \sum_{i=1}^{n} R(\tau_i | h) $$
> > This transforms the learning signal from a sparse, binary reward into a dense, continuous value (e.g., the prefix's success rate). By doing so, HiPO provides a stable, high-fidelity gradient for the crucial early steps of reasoning, as it effectively marginalizes out the variance from the diverse actions that follow the prefix.
> >
> > For stronger baseline, as detailed in the **Main Results** and **Response to W3**: HiPO outperforms DAPO across all 5 benchmarks (Average Avg@32: **54.8** vs. 53.7). Crucially, HiPO achieves this with $\approx 1/4$ of the prompt volume required by DAPO.

---

> > > ### Author Response · Authors · 2025-11-25
> > > **To reviewer DaaS (Part 3)**
> > >
> > > **Q3:**
> > >
> > > > The hint sampling uses discrete ratios [0.05-0.45]; have you ablated continuous sampling or adaptive lengths (e.g., based on intermediate reward proxies)? Could longer hints (>0.45) be viable with variance regularization to prevent collapse?
> > > >
> > >
> > > To directly address the sensitivity of the hint ratio, we conducted a new ablation study (detailed in **Section 5.4**) comparing HiPO's dynamic range against two static extremes: **Fix-Low** ($p=0.05$) and **Fix-High** ($p=0.80$).
> > >
> > > **1. Ablation Study & Why Heavy Hints Underperform**
> > >
> > > We found that a heavy hint underperform, but for a different reason than variance collapse. As shown in **Figure 5**, the **Fix-High (0.8)** setting leads to significant performance degradation. The training dynamics in **Figure 4** reveal the cause: Exploration Collapse. When the hint is too long, the task becomes trivial pattern completion. The model's tool-use turns drop, indicating it stops engaging in the complex, long-horizon reasoning required for mathematical mastery. Conversely, **Fix-Low (0.05)** maintains high entropy but fails to extend tool call turns, indicating active but ineffective exploration.
> > >
> > > **2. Discrete vs. Adaptive Sampling**
> > >
> > > We found that our discrete random sampling acts as an effective, implicit adaptation mechanism. By uniformly covering the range [0.05, 0.45], it forces the model to handle both "minimal guidance" (requiring maximal reasoning) and "strong scaffolding" within the same batch. This provides the benefits of a curriculum without the computational overhead and instability of training an auxiliary value predictor for adaptive lengths.
> > >
> > > **Q4:**
> > >
> > > > Can you provide results on stronger baselines like PPO (for direct RL comparison), DPO (to assess preference optimization alternatives), or recent hint-augmented methods such as StepHint (ICML 2025)? Specifically, how does HiPO perform relative to these on the harder benchmarks?
> > > >
> > >
> > > We prioritized comparisons based on methodological relevance and availability. Here is the rationale for excluding these specific baselines:
> > >
> > > **1. PPO (Orthogonal Contribution)**
> > >
> > > HiPO is a signal-shaping framework (curriculum), not a policy update algorithm (optimizer). It operates orthogonally to the backend optimizer. We selected GRPO as our backbone because it is the current standard for reasoning tasks (e.g., DeepSeekMath); by eliminating the value network, GRPO allows us to isolate the contribution of the *hinting mechanism* without the confounding instability of PPO's critic training.
> > >
> > > **2. DPO (Incompatible Problem Setting)**
> > >
> > > DPO is designed for preference learning and fundamentally requires paired preference data ($y_{chosen}, y_{rejected}$). In contrast, HiPO operates in the RLVR (Reinforcement Learning from Verifiable Rewards) setting, where only sparse, binary outcome rewards ($r \in \{0, 1\}$) are available. DPO cannot be applied in this standard RLVR context without constructing a separate synthetic preference dataset, which fundamentally alters the task definition.
> > >
> > > 3. StepHint (Concurrent & Unavailable)
> > > StepHint is a concurrent work (currently under review at ICLR 2026) with no publicly available code, making rigorous reproduction infeasible. Furthermore, StepHint relies on exogenous hints (from teachers or datasets), whereas HiPO's core novelty is endogenous self-hinting, which requires no external supervision.
> > >
> > > **Alternative Strong Baseline (DAPO)**
> > >
> > > To ensure rigor, we instead compared HiPO against the most relevant SOTA baseline: DAPO, which specifically targets the sparse reward problem in RLVR. As detailed in the **Main Results**, HiPO outperforms DAPO in both accuracy (Avg: **54.8** vs. 53.7) and sample efficiency (**4x less compute**).

---

### Official Review · Reviewer_Wgcs · 2025-11-01

**Soundness:** 3
**Presentation:** 3
**Contribution:** 2
**Rating:** 4
**Confidence:** 4

**Summary:**

HiPO is a novel RLVR framework designed for sparse reward reasoning tasks like mathematics. It overcomes issues in standard policy gradient methods through its "Endogenous Self-Hint" mechanism. This mechanism captures successful trajectory prefixes as on-policy hints, generating high-signal guided trajectories that transform sparse rewards into a dense, contrastive learning signal. HiPO significantly outperforms a GRPO baseline on five math benchmarks.

**Strengths:**

1. The paper is well-written and easy to follow.
2. The idea in this paper is simple yet efficient.
3. This idea can maintain the exploration property.

**Weaknesses:**

1. When facing extremely challenging tasks, the paper doesn't clarify how HiPO handles "total failure" batches where all groups have 0% success. Since hints are sourced from "Near-miss Groups" (0-50% success), an entirely "Unlearnable" batch would leave the hint pool empty, breaking the feedback loop and preventing hint-guided trajectory generation. This "cold start" scenario is unaddressed. Could you please clarify the precise mechanism for handling a mini-batch where no successful trajectories are generated (i.e., all groups are "unlearnable" with 0% success)? Does the hint-generation step simply fail for that batch, and the model must rely on standard GRPO updates until a success is stochastically found? Or is there a different mechanism to source hints (e.g., from a global buffer of past successes)?
2. A lot of hyperparameters need to be ablated and discussed: (a). Length range [0.05, 0.45] for the hint. (b). When to activate the HiPO, when $0<H_{\text{pool}} < \frac{n}{2}$.
3. The baselines are limited; please introduce recently published algorithms for comparison, e.g., DAPO.
4. The authors introduced a mechanism through a two-stage sampling process to encourage the diversity of hints.  How to measure this enhanced diversity from the proposed approaches? Any other methods?

Minors:
1. In Figure 1, the authors should clearly state what is the meaning of numbers, e.g., 0.2, 0.5, 0.7, 0.9, though I understand from the later part: the length of the hint prompt.

**Questions:**

See weakness.

---

> ### Author Response · Authors · 2025-11-25
> **To reviewer Wgcs**
>
> Thank you for your valuable feedback and address the raised weakness and questions as follows:
>
> **Response to weaknesses & questions**
>
> **W1:**
>
> > When facing extremely challenging tasks, the paper doesn't clarify how HiPO handles "total failure" batches where all groups have 0% success. Since hints are sourced from "Near-miss Groups" (0-50% success), an entirely "Unlearnable" batch would leave the hint pool empty, breaking the feedback loop and preventing hint-guided trajectory generation. This "cold start" scenario is unaddressed.
>
> > Could you please clarify the precise mechanism for handling a mini-batch where no successful trajectories are generated (i.e., all groups are "unlearnable" with 0% success)?
> >
>
> In the specific scenario where an entire mini-batch contains zero successful trajectories, the hint mechanism does not activate. By design, HiPO defaults to a standard GRPO update for that step. This fallback ensure the model learns strictly from its own on-policy experience rather than relying on potentially misleading off-policy hints.
>
> In practice, our stress test on the hardest 10% of Omni-Math problems shows how HiPO avoids getting stuck in such "cold starts." In that near-zero success environment, HiPO's strength lies in its ability to capitalize on even a few rare, stochastic successes occurring anywhere within the *global* batch. **As our results show (Appendix C, Fig. 8)**, this batch-level signal broadcasting is highly effective at finding these needles in the haystack and sharing them as hints, ensuring the learning process can continue even when most trajectories are failures. So, while a total failure batch is possible, HiPO is designed to be robust as long as successes are not *completely* impossible.
>
> **W2:**
>
> > Does the hint-generation step simply fail for that batch, and the model must rely on standard GRPO updates until a success is stochastically found? Or is there a different mechanism to source hints (e.g., from a global buffer of past successes)?
> >
>
> Yes, in the absence of successful trajectories, the method relies on standard GRPO updates. We view the discovery of the initial success as a stochastic event; the GRPO fallback serves as the exploration engine until a success is found (consistent with pass@k scaling), at which point HiPO's feedback loop is automatically activated.
>
> Regarding the use of a global buffer (past successes), we deliberately avoided this to maintain a strictly **on-policy** regime. Stored hints inevitably introduce distribution shift as the policy evolves. Our ablation study in **Section 5.5** validates this design choice: we compared HiPO against an "Off-Policy Hint" baseline (using static hints). The results showed that off-policy guidance caused performance to regress below the GRPO baseline (**Figure 5**) and led to **exploration collapse**, where the model ceased trying complex tool-use chains (**Figure 4**). This confirms that falling back to GRPO is preferable to injecting misaligned off-policy priors; HiPO ensures that guidance evolves synchronously with the agent's capabilities.
>
> **W3:**
>
> > A lot of hyperparameters need to be ablated and discussed: (a). Length range [0.05, 0.45] for the hint. (b). When to activate the HiPO, when 0 < H_pool < n/2
> >
>
> **(a) Hint Length Ablation:**
> To validate the dynamic hint range [0.05, 0.45], we conducted an ablation study (detailed in **Section 5.4**) comparing HiPO against static **Fix-Low** (0.05) and **Fix-High** (0.80) baselines. **Figure 5** demonstrates that HiPO consistently outperforms both. The training dynamics explain why: **Fix-High** leads to **mode collapse** (lowest entropy and minimal tool usage in **Figure 4**), as excessive guidance reduces the task to trivial completion. Conversely, **Fix-Low** maintains high entropy but slow to extend tool call turns, indicating active but ineffective exploration. HiPO's dynamic strategy strikes the critical balance, providing sufficient scaffolding to foster complex tool use without suppressing the exploration required for robust learning.
>
> **(b) Activation Condition** ($0 < |H_{pool}| < n/2$):
> This threshold is designed to target "boundary" examples where success is present but fragile. The lower bound ($>0$) is a logical necessity to ensure on-policy hints exist. The upper bound ($<n/2$) serves as an efficiency filter: we disable hinting for groups where the success rate already exceeds 50%. If a prompt is already "easy" for the policy, injecting hints becomes redundant. By focusing on the "near-miss" zone, HiPO directs the computational budget toward scenarios where the model is on the cusp of mastery, maximizing the marginal utility of the gradient signal.

---

> > ### Author Response · Authors · 2025-11-25
> > **To reviewer Wgcs (part 2)**
> >
> > **W4:**
> >
> > > The baselines are limited; please introduce recently published algorithms for comparison, e.g., DAPO.
> > >
> >
> > We thank the reviewer for suggesting this relevant baseline. We have conducted a rigorous comparative analysis with **DAPO** . The results are summarized below:
> >
> > Table: Performance Comparison (Avg / Maj @32)
> >
> > | Benchmark | Metric | Qwen3-8B | GRPO | DAPO | **HiPO** |
> > | --- | --- | --- | --- | --- | --- |
> > | **AIME 24** | Avg | 54.7 | 72.1 | 76.0 | **76.7** |
> > | **AIME 25** | Avg | 47.6 | 63.0 | 63.7 | **66.1** |
> > | **BRUMO 25** | Avg | 30.3 | 41.7 | **47.8** | 46.6 |
> > | **HMMT 25** | Avg | 14.0 | 28.6 | **31.4** | 30.8 |
> > | **CMIMC 25** | Avg | 37.0 | 43.5 | 49.9 | **53.8** |
> > | **Average** | Avg | 36.7 | 49.8 | 53.7 | **54.8** |
> > | **Average** | Maj | 56.0 | 57.0 | 57.8 | **61.7** |
> >
> > **1. Superior Performance at Equivalent Training Steps**
> > As shown in the tables above, HiPO consistently outperforms DAPO across the aggregate metrics. HiPO achieves a higher overall **avg@32 (54.8 vs. 53.7)** and a significantly higher **maj@32 (61.7 vs. 57.8)** across the five benchmarks. This demonstrates that even when aligned for optimization steps, HiPO provides a stronger learning signal.
> >
> > **2. Critical Advantage in Sample Efficiency**
> > Crucially, the performance gap is even more significant when considering the actual computational cost. DAPO's dynamic sampling mechanism relies on generating additional candidate batches to find non-zero rewards. In our standard configuration, this means **DAPO consumes approximately 4 times the prompt volume (total generated tokens) of HiPO per training step.**
> >
> > As detailed in our new **Appendix B** and **Figure 7**, when we normalize performance against the actual Prompt Volume, HiPO converges significantly faster and reaches a higher peak, while DAPO exhibits a long "tail" of data consumption reflecting its reliance on brute-force rejection sampling. In summary, HiPO is not only more effective in final performance but fundamentally more sample-efficient.
> >
> > **W5:**
> >
> > > The authors introduced a mechanism through a two-stage sampling process to encourage the diversity of hints. How to measure this enhanced diversity from the proposed approaches? Any other methods?
> > >
> >
> > Our two-stage sampling strategy—randomizing both the source trajectory and the hint length—is a theoretical necessity within the GRPO framework, not merely a heuristic for variety. Without this structural variance, the method suffers from "pathological symmetry." Specifically, if an entire group shares an identical fixed prefix, the sum of standardized advantages ($\sum \hat{A}_i$) is zero by definition, rendering the gradient for the hint tokens null ($\nabla J \propto \sum \hat{A}_i = 0$). By varying hint lengths, we break this symmetry, ensuring valid, non-zero learning signals that allow the model to learn from the hints effectively.
> >
> > To quantify the resulting diversity, we analyze two key metrics in **Figure 4**. First, Policy Entropy acts as a direct proxy for the diversity of the generation distribution; HiPO consistently maintains higher entropy than the baseline, indicating effective avoidance of mode collapse. Second, the steady increase in Tool-Use Turns confirms that the model is learning diverse, long-horizon reasoning paths rather than overfitting to short, trivial solutions.
> >
> > **Q1:**
> >
> > > In Figure 1, the authors should clearly state what is the meaning of numbers, e.g., 0.2, 0.5, 0.7, 0.9, though I understand from the later part: the length of the hint prompt.
> > >
> >
> > We apologize for the ambiguity and have revised the caption of Figure 1 to explicitly state that these numbers represent the hint ratio (the proportion of the total trajectory length used as the hint).

---

### Official Review · Reviewer_eFHX · 2025-11-02

**Soundness:** 3
**Presentation:** 3
**Contribution:** 4
**Rating:** 6
**Confidence:** 4

**Summary:**

The paper identifies and targets two core failure / difficulties in RLVR on long-chain mathematical reasoning tasks, namely the near-miss problem and exploration stagnation. It proposes HiPO, a self-bootstrapping framework that extracts partial prefixes (“self-hints”) from the few successful trajectories within a batch and reuses them to regenerate a new, more informative batch on the same prompts. By replacing low- or zero-signal groups with hint-augmented ones, HiPO effectively densifies the reward signal and makes GRPO-style optimization work even when success is rare. Experiments on recent difficult math benchmarks (CMIMC 25, BRUMO 25, AIME 24/25) show consistent improvements over GRPO/DAPO-style baselines.

**Strengths:**

- This paper is techinically simple but effectively improves the sampling efficiency of RLVR in difficult scenarios where the reward is very sparse.

- The proposed method is practically plug-and-play for GRPO-like group-based pipelines, and does not need expernal teachers' hints.

- Empirical results on recent, difficult math benchmarks show clear gains.

**Weaknesses:**

- The proposed HiPO "amplifies" the rare successful runs. However, on very hard tasks or early in training where only very rare successful runs can be sampled, HiPO may cause "over-exploitation" of a small set of successful runs. The authors are recommended to discuss this possibility in the paper.

- Comparison is mainly against GRPO/DAPO-like baselines; it would be important to see how HiPO fares against stronger exploration- or entropy-aware RL variants, or against pipelines that inject external hints. This would clarify whether “self-hint” is a better source of guidance than existing teacher-style hints, or just a cheaper one.

- Some key designs (e.g., the ratio of the original null-signal group samples that are replaced by the hint samples) lack ablations.

**Questions:**

All experiments stay in math / verifiable QA style tasks, which are the friendliest setting for RLVR because of binary, automatic reward. As the authors claim the value of the proposed HiPO for general RLVR, how does the method work on tasks where rewards are delayed, noisy, or non-binary (code, tool-using agents, long dialogues)?

---

> ### Author Response · Authors · 2025-11-25
> **To reviewer eFHX**
>
> Thank you for your valuable feedback and address the raised weakness and questions as follows:
>
> **W1&2:**
>
> > The proposed HiPO "amplifies" the rare successful runs. However, on very hard tasks or early in training where only very rare successful runs can be sampled, HiPO may cause "over-exploitation" of a small set of successful runs. The authors are recommended to discuss this possibility in the paper. Some key designs (e.g., the ratio of the original null-signal group samples that are replaced by the hint samples) lack ablations.
> >
>
> **Mitigating Over-Exploitation (Ablation Study)**
>
> We explicitly investigated over-exploitation by comparing HiPO against **Fix-High** ($p=0.8$) and **Fix-Low** ($p=0.05$) in **Section 5.4**.
>
> - **Fix-High:** Confirmed the reviewer's concern. Excessive guidance caused mode collapse (lowest entropy/tool-use in **Figure 4**), proving that static long hints lead to overfitting.
> - **Fix-Low:** Maintained entropy but slow to extend reasoning chains, indicating ineffective exploration.
> - **HiPO:** Our dynamic strategy strikes the optimal balance. It maintains high entropy (like Fix-Low) while achieving the highest tool-use turns, proving that the model learns robust reasoning rather than memorizing rare artifacts.
>
> **Proof of Generalization**
>
> Further evidence against over-exploitation is found in the hardest 10% of the Omni-Math dataset (**Appendix C**). Despite training on extremely rare successes, the model consistently improves on AIME 2024/2025, confirming that HiPO extracts transferable logic.
>
> **W3:**
>
> > Comparison is mainly against GRPO/DAPO-like baselines; it would be important to see how HiPO fares against stronger exploration- or entropy-aware RL variants, or against pipelines that inject external hints. This would clarify whether “self-hint” is a better source of guidance than existing teacher-style hints, or just a cheaper one.
> >
> **Comparison with strong DAPO baseline**
>
> To broaden our evaluation, we incorporated DAPO, a strong RLVR baseline that uses dynamic sampling to tackle sparse rewards. As shown below, HiPO consistently outperforms DAPO in aggregate metrics.
>
> Table: Performance Comparison (Avg / Maj @32)
>
> | Benchmark | Metric | Qwen3-8B | GRPO | DAPO | **HiPO** |
> | --- | --- | --- | --- | --- | --- |
> | **AIME 24** | Avg | 54.7 | 72.1 | 76.0 | **76.7** |
> | **AIME 25** | Avg | 47.6 | 63.0 | 63.7 | **66.1** |
> | **BRUMO 25** | Avg | 30.3 | 41.7 | **47.8** | 46.6 |
> | **HMMT 25** | Avg | 14.0 | 28.6 | **31.4** | 30.8 |
> | **CMIMC 25** | Avg | 37.0 | 43.5 | 49.9 | **53.8** |
> | **Average** | **Avg** | 36.7 | 49.8 | 53.7 | **54.8** |
> | **Average** | **Maj** | 56.0 | 57.0 | 57.8 | **61.7** |
>
> While the performance gap is consistent, the efficiency gap is critical. DAPO relies on dynamic sampling to locate non-zero rewards, consuming approximately $4\times$ the prompt volume of HiPO per update step. As detailed in **Appendix B**, HiPO achieves superior convergence by recycling rare signals rather than resampling for them, offering a fundamentally more scalable paradigm.
>
> **Comparison with off-policy Hint baseline**
>
> We compared HiPO against an Off-Policy Hint baseline (detailed in **Section 5.5**), where hints were derived from successful trajectories of the frozen base model (simulating static/synthetic guidance). The results were negative:
>
> 1. Performance Regression: The Off-Policy baseline underperformed not only HiPO but also the standard GRPO baseline (see **Figure 5**).
>
> 2. Exploration Collapse: As shown in **Figure 4**, the model's tool-use turns collapsed to near zero. Relying on static, exogenous hints caused the policy to overfit to simplistic paths ("distribution drift"), effectively shutting down the exploration of complex, multi-step reasoning chains.
>
> This validates our design choice: it is better to fallback to standard GRPO during total failure than to inject potentially misaligned off-policy priors. HiPO’s strict adherence to endogenous, on-policy hints ensures that the guidance evolves alongside the agent, maintaining the integrity of the exploration process.

---

> > ### Author Response · Authors · 2025-11-25
> > **To reviewer eFHX (Part 2)**
> >
> > **Q1:**
> >
> > > All experiments stay in math / verifiable QA style tasks, which are the friendliest setting for RLVR because of binary, automatic reward. As the authors claim the value of the proposed HiPO for general RLVR, how does the method work on tasks where rewards are delayed, noisy, or non-binary (code, tool-using agents, long dialogues)?
> > >
> >
> > We argue that HiPO is structurally well-suited for general RLVR tasks (Code, Agents) precisely because it addresses the core challenges usually associated with them.
> >
> > **1. Handling Delayed Rewards (The Credit Assignment Problem)**
> >
> > Long-horizon Math (our setting) and Code Generation are structurally isomorphic: both involve a long sequence of actions with a single reward at the very end. HiPO solves the resulting credit assignment problem via Monte Carlo Value Estimation. By enforcing a shared hint $h$ and averaging outcomes over $n$ futures, HiPO computes:
> > $$ V(h) \approx \frac{1}{n} \sum_{i=1}^{n} r(\tau_i | h) $$
> > This transforms a "delayed" binary outcome into an immediate, dense value estimate for the prefix, bridging the gap between early planning steps and the final reward.
> >
> > **2. Handling Non-Binary/Noisy Rewards**
> >
> > HiPO handles non-binary rewards natively. The formula above works identically for scalar rewards (e.g., passing 80% of unit tests in Code Gen) as it does for binary ones. The averaging process naturally suppresses noise, providing a stable signal closer to the true expected value of the reasoning path.

---

### Author Response · Authors · 2025-11-27
**Global Response**

We sincerely thank all reviewers for their time and insightful comments, which have helped us significantly improve our manuscript. We are encouraged that the reviewers found our work:

- **Simplicity and Efficiency**: HiPO is concise yet highly effective, improving RLVR sampling efficiency in sparse-reward scenarios (**Reviewer eFHX**, **Reviewer DaaS, Reviewer Wgcs**). The method is plug-and-play for GRPO-like pipelines and does not require external teacher hints (**Reviewer eFHX, Reviewer DaaS**).
- **Targeting Core RL Challenges**: HiPO resolves critical RL issues: signal collapse and credit misassignment by transforming rare successes into reusable on-policy guidance (**Reviewer DaaS, Reviewer Hyap**). This design supports exploration without external guidance (**Reviewer Wgcs**, **Reviewer Hyap**).
- **Validated Anti-Stagnation**: HiPO effectively combats stagnation, improving learnable group proportions and sustaining higher entropy and tool integrate rate than baselines (**Reviewer DaaS**).

Additionally, all reviewers acknowledged the empirical benefits of our method in comparison to baselines on competition-level math benchmarks.

To address reviewers’ valuable feedback, we have made substantial revisions to our paper. The main changes are summarized below:

- **Stronger DAPO baseline**: We have added a new set of experiments comparing our method with DAPO. The results and detailed data efficiency discussion in **Table 1** and **Appendix B** show that our method not only achieves better performance but also demonstrates superior data efficiency compared to this baseline.
- **Ablation Study of Hint Ratios**: We added an ablation study on the Hint ratio, testing fix-Low (ratio=0.05, weak guidance) and fix-High (ratio=0.80, excessive guidance). **Section 5.4** discusses how our method balances exploration and exploitation.
- **Ablation Study of Off-Policy Hint**: We added a comparison with off-policy hinting. The experiments and discussion in **Section 5.5** show that the off-policy method heavily depends on the quality of the hints, and using hints extracted from the base model leads to exploration stagnation.
- **Robustness to Ultra-Sparse Rewards**: We ran HiPO on the hardest 10% of the Omni-Math dataset, simulating extreme cases with very sparse positive samples. **Appendix C** discusses the robustness of our method in such scenarios.
- **Other Evaluation**: We add evaluation on the OOD HumanEval task. **Appendix E** discusses its generalization on this task.

We have carefully addressed each reviewer’s comments in detail below. We believe that our revisions and responses have adequately addressed all concerns, and we sincerely hope that the reviewers will find the updated manuscript significantly strengthened.

---

### Meta-Review · Area_Chair_gMLR · 2026-01-06

**Summary:**

HiPO introduces endogenous self‑hinting for RLVR: when the model occasionally produces a successful trajectory, HiPO extracts the initial correct prefix and reuses it as an on‑policy “hint” to regenerate guided trajectories for groups that otherwise receive no reward signal. This transforms sparse, binary rewards into dense contrastive learning signals, combats exploration stagnation, and improves credit assignment by estimating prefix value via Monte Carlo rollouts. Across five challenging math‑reasoning benchmarks, HiPO consistently outperforms GRPO and DAPO while being more sample‑efficient, demonstrating a scalable path for LLMs to bootstrap from their own rare successes.

Reviewers appreciated the simple but effective method, the novelty of the self-hint paradigm for on-policy guidance, writing, and empirical gains.

They also raised several concerns:
- Cold-start / total failure batches: what happens if no successes exist?
- Risk of over-exploitation of rare successes, and resulting mode-collapse.
- Limited baselines (requested DAPO, PPO, STaR etc.)
- The framework has several hyperparameters (hint ratio, activation threshold).

**Reviewer Concerns:**

Addressed:
- Cold-start / total failure batches: what happens if no successes exist? Authors report that HiPO falls back to GRPO in this case.
- Risk of over-exploitation of rare successes, and resulting mode-collapse. Authors pointed to ablations ("Fix-Low", "Fix-High") showing that HiPO avoids collapse through dynamic hinting that balances exploration / exploitation.


Partially addressed / unaddressed:
- Limited baselines (requested entropy-aware baselines, DAPO, PPO, STaR etc.). Authors added DAPO comparisons, and justified excluding PPO and STaR. But still no comparisons to explicit exploration- or entropy-aware RL variants, or against pipelines that inject external hints.
- The framework has several hyperparameters (hint ratio, activation threshold). Here, the author response attempts to justify the choices in the paper, but it still appears unclear what the sensitivity of the method is to these choices.

**Reviewer Scores:**

eFHX: 6->6

Wgcs: 4->6

DaaS: 4->6

HyaP: 6->6

While borderline, I think the novelty of the idea and the strength of empirical results push this one over the bar.

---

### Decision · Program_Chairs · 2026-01-26

Accept (Poster)